# A conserved membrane curvature-generating protein is crucial for autophagosome formation in fission yeast

Ning Wang [1], Yoko Shibata[1], Joao A. Paulo [2], Steven P. Gygi [2] & Tom A. Rapoport [1] ✉

Organelles are shaped by curvature-generating proteins, which include the reticulons and REEPs that are involved in forming the endoplasmic reticulum (ER). A conserved REEP subfamily differs from the ER-shaping REEPs in abundance and membrane topology and has unidentified functions. Here, we show that Rop1, the single member of this family in the fission yeast *Schizosaccharomyces pombe*, is crucial for the macroautophagy of organelles and cytosolic proteins. Rop1 is needed for the formation of phagophores, cup-like structures consisting of two closely apposed membrane sheets that encapsulate cargo. It is recruited at early stages to phagophores and is required for their maturation into autophagosomes. Rop1 function relies on its ability to generate high membrane curvature and on its colocalization with the autophagy component Atg2 that is thought to reside at the phagophore rim. We propose that Rop1 facilitates the formation and growth of the double-membrane structure of the autophagosome.

The ER consists of narrow tubules and planar sheets of closely apposed membranes. The high membrane curvature of ER tubules and sheet edges is generated by two evolutionarily conserved protein families, the reticulons (Rtns) and the REEP5 branch of the REEPs (REEP5,6 in mammals and Yop1 in yeast)[1–4]. The REEP5 subfamily and Rtns have redundant functions in shaping the ER. They are abundant in all eukaryotic cells and share a similar structural organization, consisting of four transmembrane segments (TMs) and an amphipathic helix (APH). These features are required to generate high membrane curvature[5].

The REEP family contains another branch (REEP1-4 in mammals) whose function is unclear. Although they have been proposed to participate in ER formation[6], they likely play a limited role, as they are much less abundant than the reticulons and REEP5/Yop1. They are predicted to contain a C-terminal APH and three preceding TMs that are superimposable with the last three TMs of REEP5 subfamily members in Alphafold predicted models[7] (Fig. 1a; Supplementary Fig. 1a, b). REEP1 proteins are found in most eukaryotic organisms and cells (Supplementary Fig. 1c). In vertebrates, the four members of this subfamily are differentially expressed in distinct cell types. In fission yeast such as *Schizosaccharomyces pombe*, there is a single uncharacterized ortholog that we have named Rop1 (Reep one protein 1; *SPBC30D10.09c*). The only eukaryotes that seem to lack obvious REEP1 orthologs are budding yeasts. These organisms instead possess an unrelated protein of similar membrane topology, Atg40, which serves as a receptor for autophagy of the ER (ERphagy)[8–10]. Because Atg40 is not found outside budding yeasts, we wondered whether REEP1 proteins play a role in autophagy in other organisms.

Macroautophagy (herein called autophagy) can be induced by starvation, the damage or overabundance of organelles, or the accumulation of misfolded proteins. Autophagy begins with the formation of a membrane sheet, called the phagophore or isolation membrane, which consists of two closely apposed phospholipid bilayers connected by a highly curved rim[11–16] (Supplementary Fig. 1d). The membrane sheet expands and bends, generating a cup-like structure, and eventually closes on itself to form an autophagosome, in which the two membranes are no longer connected. During closure, the autophagosome enwraps cytosolic components or organelles, either by

[1]Howard Hughes Medical Institute and Department of Cell Biology, Harvard Medical School, 240 Longwood Avenue, Boston, MA 02115, USA. [2]Department of Cell Biology, Harvard Medical School, 240 Longwood Avenue, Boston, MA 02115, USA. ✉e-mail: tom_rapoport@hms.harvard.edu

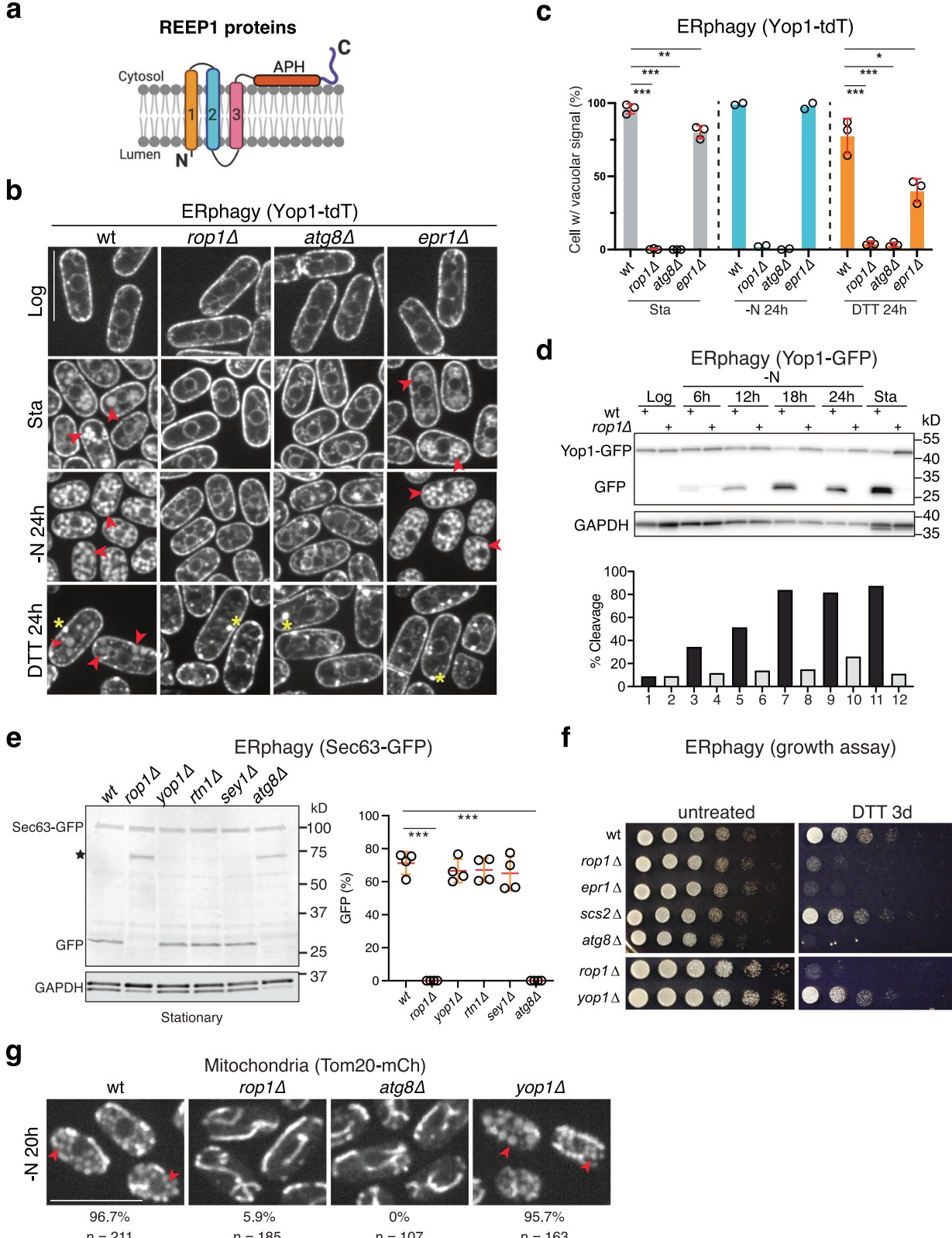

**a** REEP1 proteins

**b** ERphagy (Yop1-tdT)

**c** ERphagy (Yop1-tdT)

**d** ERphagy (Yop1-GFP)

**e** ERphagy (Sec63-GFP)

**f** ERphagy (growth assay)

**g** Mitochondria (Tom20-mCh)

random encapsulation (bulk autophagy) or by binding to specific receptors on the cargo (selective autophagy). The outer membrane of the autophagosome subsequently fuses with a lysosome (or vacuole in yeast), and both the inner membrane and the captured content are eventually degraded (Supplementary Fig. 1d).

Autophagy requires a number of conserved core components[11,17,18] (Supplementary Fig. 1d). Some of these are involved in the attachment

of a ubiquitin-like protein (Atg8 in yeast and LC3/GABARAP in mammals) to the lipid phosphatidylethanolamine in the phagophore membrane. Atg8 lipidation is critical for phagophore biogenesis and for the completion of autophagy. However, how exactly the phagophore forms remains unclear. The rim is highly curved (diameter of 15–17 nm in yeast)[19], although this region's intermembrane distance is somewhat larger than in the rest of the phagophore[15,16]. Such high

**Fig. 1 | *S. pombe* Rop1 is required for autophagy of organelles. a** Membrane topology of Rop1 and other REEP1 family proteins. Transmembrane (TM) segments are numbered. APH, amphipathic helix. **b** ERphagy in *S. pombe* was tested with a tdTomato fusion of the ER protein Yop1 (Yop1-tdT) expressed at endogenous levels in wild-type (wt) cells or cells lacking the indicated proteins. Cells were imaged by fluorescence microscopy at logarithmic (Log) or stationary (Sta) growth phase, after nitrogen starvation for 24 h (-N 24 h), or after treatment with DTT for 24 h (DTT 24 h). Scale bar, 5 μm. Red arrowheads point to vacuoles and yellow stars to Yop1 aggregates. **c** Quantification of experiments in (**b**). Shown is the percentage of cells with vacuolar fluorescence. The circles refer to three individual experiments. Also shown are means and standard deviations (SD). *, **, and *** indicate significant differences with *p*-values < 0.1, 0.01, and 0.001, calculated from two-tailed Student's *t* tests. The exact *p*-values are listed in the Source Data file. **d** Cleavage of GFP-

tagged Yop1 (Yop1-GFP) was tested in wt or *rop1Δ* cells grown at Log or Sta phase or deprived of nitrogen (-N) for different time periods. Lysates were analyzed by SDS-PAGE and immunoblotting with GFP antibodies. Blotting with GAPDH antibodies served as loading control. The lower panel shows the percentage of Yop1-GFP cleavage. **e** As in (**d**), but with the GFP-tagged ER marker Sec63 (Sec63-GFP) tested in wt and mutant strains. The star indicates a cleavage product not caused by ERphagy. The right panel shows quantification of Sec63-GFP cleavage in four experiments, performed as in (**c**). **f** wt or mutant cells were treated with DTT for 3 days and plated after serial dilution. Controls were performed with untreated cells. **g** As in (**b**), but for the mitochondrial protein Tom20 tagged with mCherry (Tom20-mCh). The percentage of cells with vacuolar fluorescence and the number of cells analyzed are given below the images. Scale bar, 10 μm.

membrane curvature is energetically unfavorable and requires proteins for stabilization[13]. Some autophagy factors such as Atg2 and Atg18 localize to the rim[20,21], but none contain obvious domains that could stabilize such high curvature. Atg2 is thought to form a channel that allows lipids to flow from the ER into the growing phagophore[22,23]. Because high curvature regions of the ER are often juxtaposed to the phagophore rim[15,16], it is possible that lipid transfer occurs at these contact sites.

Here, we take advantage of the fact that Rop1 is the single member of the REEP1 subfamily in *S. pombe*. We show that Rop1 plays a crucial role in all forms of autophagy and is not just an autophagy receptor like Atg40 in *S. cerevisiae*. We provide evidence that Rop1 stabilizes high membrane-curvature regions to allow the formation and growth of phagophores.

## Results

### Rop1 is important for all forms of autophagy in *S. pombe*

We first investigated whether Rop1 participates in ERphagy in *S. pombe*. We fused the ER protein Yop1 to the fluorescent proteins tdTomato (Yop1-tdT) or GFP (Yop1-GFP) and tested whether these proteins move from the ER into vacuoles under autophagy-inducing conditions. In logarithmically growing wild-type cells, Yop1-tdT and Yop1-GFP localized to the cortical ER as expected (Fig. 1b). In contrast, they accumulated in vacuoles after ERphagy was induced, either by growing the cells to stationary phase or by depriving them of nitrogen (Fig. 1b, c; Supplementary Fig. 2a). Triggering protein misfolding in the ER with dithiothreitol (DTT) also caused vacuolar accumulation, although it also triggered membrane protein aggregation[24] (Fig. 1b, c). Strikingly, no vacuolar accumulation of these ERphagy markers was seen in cells lacking Rop1 (*rop1Δ*) (Fig. 1b, c; Supplementary Fig. 2a), indicating a pronounced inhibition of autophagy. This defect was as strong as observed after the deletion of Atg8 (*atg8Δ*). The only known ERphagy receptor in *S. pombe* is Epr1[24], but its absence (*epr1Δ*) reduced autophagy only after DTT treatment (Fig. 1b, c).

We further assessed ERphagy by monitoring the proteolytic cleavage of Yop1-GFP in vacuoles[24,25]. In cells grown to stationary phase or starved of nitrogen, the Yop1 portion of the fusion protein was digested while the GFP moiety resisted proteolysis (Fig. 1d). No processing of Yop1-GFP occurred in *rop1Δ* cells (Fig. 1d), demonstrating a complete block of ERphagy. Cleavage of Yop1-GFP was also blocked in *rop1Δ* cells after inducing ERphagy with DTT or tunicamycin (Supplementary Fig. 2b). These defects were again as pronounced as observed in *atg8Δ* cells (Supplementary Fig. 2b). Similar results were obtained in stationary or nitrogen-starved cells using Yop1-mRFP or Rtn1-mRFP as ERphagy markers (Supplementary Fig. 2c). Because these markers localize to high-curvature regions of the ER, we tested whether the autophagy of bulk ER is also affected by Rop1. Indeed, a GFP-fusion of the general ER marker Sec63 (Sec63-GFP) was cleaved into GFP in stationary wild-type cells, but not in *rop1Δ* or *atg8Δ* strains (Fig. 1e). Although Rop1 is homologous to Yop1 (Supplementary Fig. 1b,c), deletion of Yop1 (*yop1Δ*) did not affect ERphagy (Fig. 1e), as observed

in *S. cerevisiae*[25]. The absence of the ER shaping proteins Rtn1 (*rtn1Δ*) or Sey1 (*sey1Δ*) also did not inhibit autophagy (Fig. 1e). As in *S. cerevisiae*[25], the single deletions of the ER shaping proteins had little effect on ER morphology (Supplementary Fig. 2d), although a slight defect was seen in *rtn1Δ* cells. The ER was also normal in *rop1Δ* cells (Supplementary Fig. 2d), indicating that the ERphagy defect is not caused indirectly by ER morphology changes. Taken together, these results show that Rop1, but not the ER shaping proteins Yop1, Rtn1, or Sey1, is required for ERphagy.

Consistent with a crucial role for Rop1 in ERphagy, *rop1Δ* cells did not survive extended DTT treatment, similarly to *atg8Δ* or *epr1Δ* cells, whereas growth of wild-type and *yop1Δ* cells was unaffected (Fig. 1f). The role of Rop1 in ERphagy is also conserved in the fission yeast *Schizosaccharomyces japonicus;* deletion of the single Rop1 ortholog (*SJAG_05289*) prevented the accumulation of Rtn1 fused to mNeon-Green (Rtn1-mNG) inside vacuoles after inducing ERphagy with the Tor inhibitor rapamycin (Supplementary Fig. 2e). Collectively, these results establish that Rop1 is essential for ERphagy.

Rop1 is also critical for autophagy of mitochondria (mitophagy) and peroxisomes (pexophagy). A fusion of the mitochondrial membrane protein Tom20 with mCherry (Tom20-mCh) accumulated inside vacuoles in nitrogen-starved wild-type or *yop1Δ* cells, whereas this accumulation was greatly reduced in *rop1Δ* or *atg8Δ* cells (Fig. 1g). The absence of Rop1 also impaired the vacuolar cleavage of Tom20-mCh (Supplementary Fig. 2f). Likewise, both the vacuolar accumulation and proteolytic cleavage of the peroxisomal protein Pex11 fused to mCherry (Pex11-mCh) were reduced in *rop1Δ* cells (Supplementary Fig. 2g, h).

Finally, we found Rop1 to be important for bulk autophagy. The cytosolic marker proteins YFP and Tdh1-mCh accumulated inside vacuoles of nitrogen-starved or stationary wild-type cells, but not in *rop1Δ* cells (Fig. 2a, b). Vacuolar cleavage of the cytosolic marker proteins Hsc1-GFP, Pyk1-mCh, and Pgk1-mCh was also inhibited in *rop1Δ* cells under various autophagy-inducing conditions (Fig. 2c; Supplementary Fig. 3a), although not as completely as in *atg8Δ* cells. An autophagy defect was also seen in *rop1Δ* cells with proteins that are both cytosolic and bound to the ER (Sf3b-mCh and Sec24-GFP) (Supplementary Fig. 3b, c). We also tested the effect of Rop1 deletion on the turnover of Atg8 itself (Supplementary Fig. 1d). In wild-type cells, mEGFP-Atg8 accumulated inside vacuoles in stationary or nitrogen starved cells (Supplementary Fig. 3d, e) and was proteolytically cleaved in stationary phase cells (Fig. 3f). In *rop1Δ* cells, however, vacuolar accumulation and proteolytic cleavage were considerably reduced.

Quantitative isobaric tag-based proteomics showed that loss of Rop1 globally affects autophagy. In starved cells, the absence of Rop1 caused the accumulation of various proteins, whereas no significant changes occurred in the absence of Yop1 (Fig. 2d). A large number of proteins was also stabilized in cells lacking Atg8 or Atg2, as expected. Rop1 and Atg2 seem to have a less pronounced effect than Atg8, but many of the accumulating proteins were the same in all three deletion strains (Fig. 2d, e, g), consistent with these proteins acting in the same

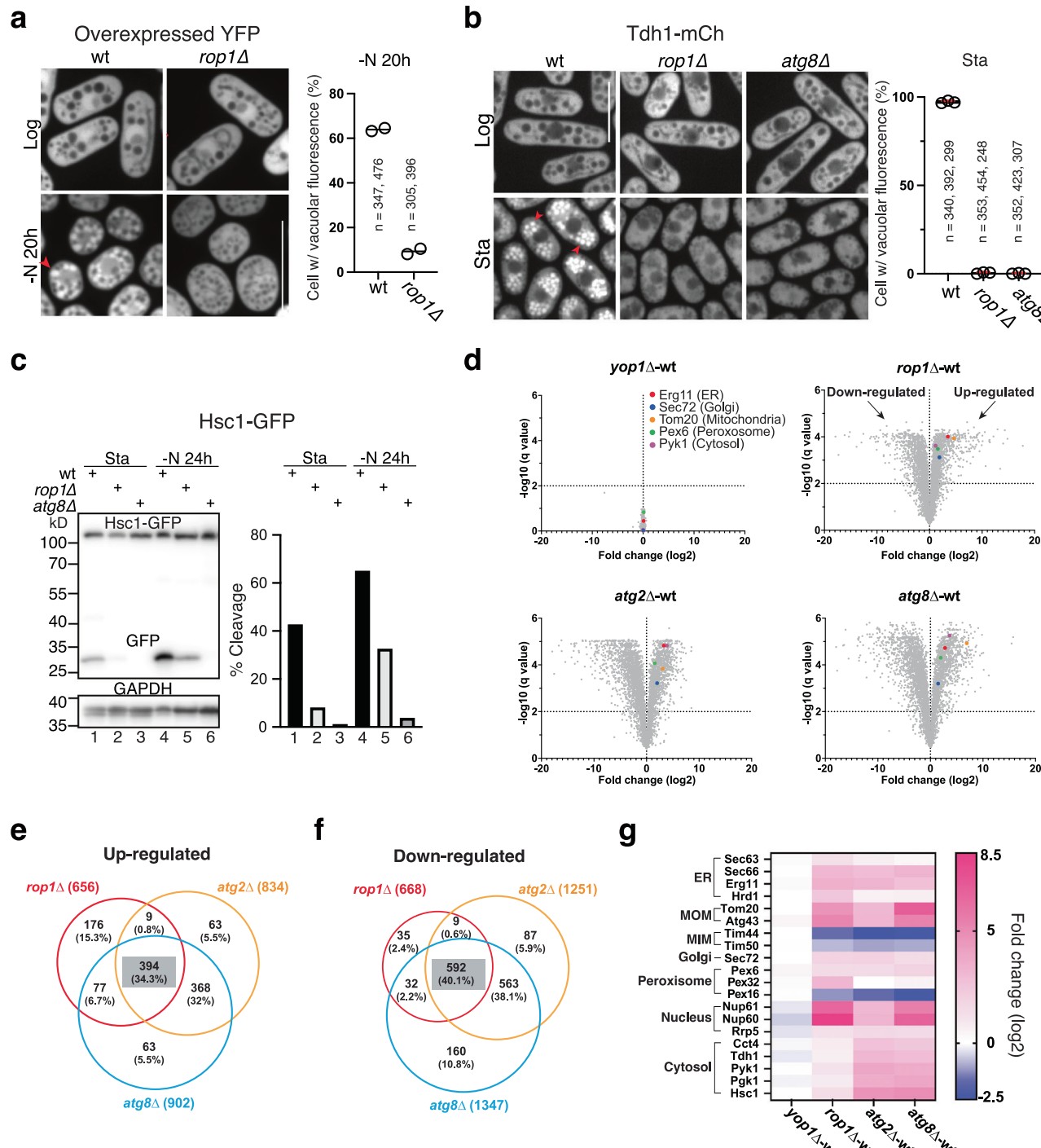

**Fig. 2 | Rop1 is crucial for bulk autophagy in *S. pombe*. a** Bulk autophagy was tested with cytosolically expressed YFP in wild-type (wt) or *rop1Δ* cells by fluorescence microscopy. The cells were analyzed in Log phase or after nitrogen starvation for 20 h (-N 20 h). Scale bar, 5 μm. The right panel shows the percentage of cells with vacuolar fluorescence in two experiments (circles) together with the means. **b** As in (**a**), but for a mCherry fusion of the cytosolic protein Tdh1 (Tdh1-mCh) analyzed in Log or stationary (Sta) phase (three experiments). Shown are means and SD. **c** Cleavage of the GFP-tagged cytosolic protein Hsc1 (Hsc1-GFP) was tested in wt or mutant cells grown at Sta phase or deprived of nitrogen for 24 h. The right panel shows quantification of Hsc1-GFP cleavage. **d** The abundance of proteins in *yop1Δ*, *rop1Δ*, *atg2Δ*, and *atg8Δ* cells was compared with that in wt cells by quantitative

isobaric tag-based proteomics. Each point in the volcano plot represents a protein for which the ratio of its abundance in the mutant and in wt cells is given, as well as a measure of statistical significance (*q*-value), derived from a multiple non-parametric *t* test based on three biological replicates. Some proteins upregulated in all autophagy mutants are highlighted in color. **e** Venn diagram of the number of proteins up-regulated in different autophagy mutants. Note that many proteins are upregulated in all three mutants. **f** As in (**e**), but for proteins that are down-regulated. **g** Examples of proteins that are up- or down-regulated in the different mutants (see scale on the right). MOM, MIM, outer and inner mitochondrial membrane.

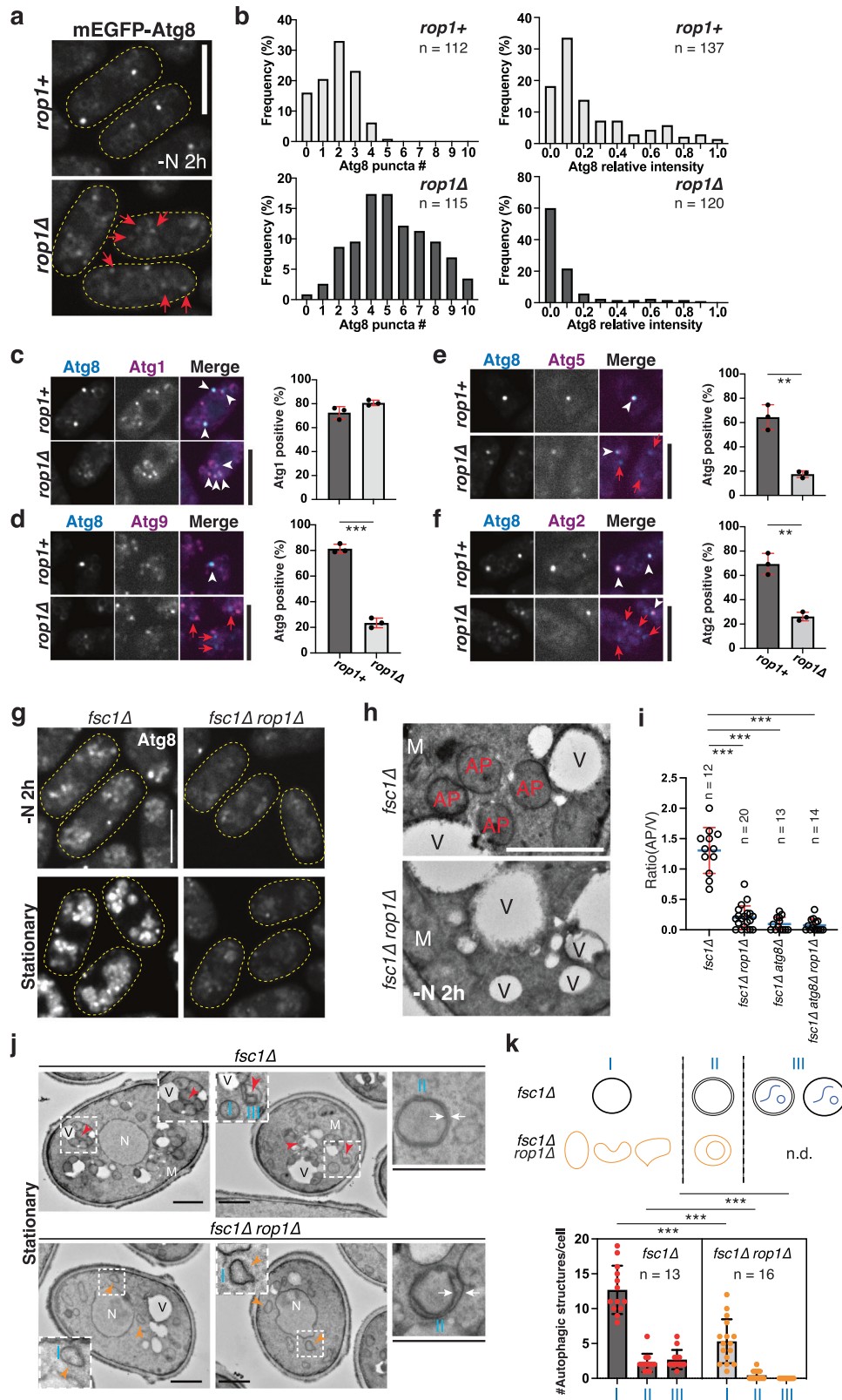

pathway. Surprisingly, the levels of many proteins decreased (Fig. 2f), perhaps as a compensatory response to the increase of other proteins that are normally degraded by autophagy.

Rop1's role in autophagy is also supported by thin-section electron microscopy (EM) (Supplementary Fig. 4a–d). In nitrogen-starved wild-type or *yop1Δ* cells, many vacuoles contained electron-dense material that likely corresponds to residual internal membranes of

autophagosomes (Supplementary Fig. 4a, b). In contrast, vacuoles were empty in *rop1Δ* and *atg8Δ* cells (Supplementary Fig. 4a, b). Similar results were obtained with cells in the stationary phase (Supplementary Fig. 4c, d). The number of vacuoles in *rop1Δ* cells was also more variable than in wild-type cells (Supplementary Fig. 4b) and many were smaller (yellow arrows in Supplementary Fig. 4a), an observation that was confirmed by light microscopy (Supplementary Fig. 4e). The

**Fig. 3 | Rop1 is required for autophagosome formation. a** Atg8 was tagged with mEGFP (mEGFP-Atg8) and expressed at endogenous levels in *rop1⁺* or *rop1Δ* cells. Cells were imaged after 2 h of nitrogen starvation (-N 2 h). The boundaries of cells are indicated by dashed lines. Dim Atg8 punctae in *rop1Δ* cells are indicated by red arrows. Scale bar, 5 µm. **b** Number of mEGFP-Atg8 punctae per cell (left) and their intensity (right), expressed relative to the brightest Atg8 punctum in *rop1⁺* cells (right). *n*, number of analyzed cells (left) or punctae (right). **c** tdTomato-tagged Atg1 was visualized together with mEGFP-Atg8 in *rop1⁺* or *rop1Δ* cells after 2 h of nitrogen starvation. The right panel shows quantification of three experiments (mean and SD). Scale bar, 5 µm. **d** As in (**c**), but with tdTomato-tagged Atg9. *** indicates a significant difference with *p*-value 4.1 × 10⁻⁵, calculated from two-tailed Student's *t* tests. **e** As in (**c**), but with tdTomato-tagged Atg5. *p*-value: 0.0015. **f** As in (**c**), but with tdTomato-tagged Atg2. *p*-value: 0.0014. **g** As in (**a**), but with *fsc1Δ* or *fsc1Δ rop1Δ* cells. The cells were nitrogen-starved for 2 h or in stationary growth phase. **h** *fsc1Δ* or *fsc1Δ rop1Δ* cells were starved for 2 h and analyzed by TEM. M mitochondrion, V vacuole, AP autophagosome. Scale bar, 500 nm. **i** Ratio of number of autophagosomes to vacuoles per section of a cell, determined in an experiment as in (**h**). Cells lacking Atg8 were also analyzed (for images, see Supplementary Fig. 5e, f). *n*, number of cells analyzed. Also shown are means and SD. *** indicates significant differences with *p*-values < 0.001, calculated from two-tailed Student's *t* tests. The exact *p*-values are listed in the Source Data file. **j** As in (**h**), but for stationary phase cells. N nucleus. Red arrowheads indicate autophagosomes with enclosed substrate, and orange arrowheads point to aberrant autophagic structures. White arrows highlight the two membranes of autophagosomes. The regions in the squares are magnified in the insets. Scale bars, 1 µm. **k** Quantification of autophagic structures in (**j**), performed as in (**i**). The scheme shows three categories of structures (I, II, III) counted. n.d., not detected. The dots in the graph give the numbers of individual structures in these categories. The bars show means and SD. *n*, number of cells analyzed.

reason for these changes is unclear, but they do not seem to be caused by gross cellular alterations, as the localization of markers of different compartments of the secretory pathway remained unaffected (Supplementary Fig. 4f).

Although Rop1 has a similar membrane topology as Atg40 in *S. cerevisiae*, it is not a functional homolog. Atg40 serves as a cargo receptor for ERphagy and contains a typical Atg8-interacting motif (AIM) that binds Atg8 in vitro and is required for the receptor's function in ERphagy[8–10]. Although Rop1 contains a potential AIM in its C-terminal segment, ablation of this motif had little effect on autophagy: cell survival in DTT (Supplementary Fig. 5a) and vacuolar accumulation of Tdh1 in the stationary phase (Supplementary Fig. 5b) were essentially normal, and only a moderate decrease was seen in vacuolar cleavage assays (see below). Neither wild-type Rop1 nor an AIM mutant bound Atg8 in vitro (Supplementary Fig. 5c). *S. cerevisiae* Atg40 expressed from the endogenous Rop1 promoter also failed to complement the growth defect of the *S. pombe rop1* mutant on DTT (Supplementary Fig. 5d). Taken together, our results indicate that Rop1 is fundamentally involved in all forms of autophagy and is not simply an ERphagy receptor.

## A crucial role for Rop1 in phagophore expansion and autophagosome formation

To understand the function of Rop1 in autophagy, we first compared the localization of various autophagy factors in wild-type versus *rop1Δ* cells under starvation conditions. Using Atg8 fused to mEGFP (mEGFP-Atg8) as a marker of autophagosomes, we noted that most wild-type cells contained two or three mEGFP-Atg8 punctae (Fig. 3a, b) that likely represent nascent phagophores. In the absence of Rop1, the number of punctae increased while their brightness decreased (Fig. 3a, b), as previously observed for *atg2Δ*, *atg18bΔ*, and *atg18cΔ* mutants[26]. In wild-type cells, the autophagy components Atg1 (the earliest autophagy marker[27]; see Supplementary Fig. 1d), Atg9 (a multi-spanning membrane protein of autophagosomes)[28], Atg5 (a component of the ubiquitin-like conjugation machinery)[29], and Atg2 (a phospholipid transfer protein)[22,23] all colocalized with mEGFP-Atg8 (Fig. 3c–f). In contrast, most of the mEGFP-Atg8 punctae in *rop1Δ* cells still contained Atg1 (Fig. 3c) but showed reduced recruitment of Atg9, Atg5, and Atg2 (Fig. 3d–f). Thus, the earliest autophagic structures can still form in the absence of Rop1, but phagophore expansion and autophagosome formation are delayed or stalled.

To further test whether Rop1 affects autophagosome formation, we prevented autophagosome fusion with the vacuole by deleting the fusion factor Fsc1[26]. After nitrogen starvation or in the stationary phase, the number of mEGFP-Atg8 punctae increased in *fsc1Δ* cells (Fig. 3g), as reported previously[26]. This accumulation was almost entirely abolished when Rop1 was also absent (*fsc1Δ rop1Δ*) (Fig. 3g). By thin-section EM, numerous spherical structures tethered to vacuoles were apparent in nitrogen-starved or stationary *fsc1Δ* cells (Fig. 3h–k).

These structures correspond to autophagosomes because they contain engulfed material (red arrows in Fig. 3j) and are not observed in the absence of Atg8 (Fig. 3i; Supplementary Fig. 5e, f). The two surrounding membranes were often too close to be distinguished, as reported in the literature[19], but occasionally they were sufficiently separated (Fig. 3j). The number of autophagosomes was significantly reduced in *fsc1Δ rop1Δ* cells and the remaining ones were irregularly shaped (orange arrows in Fig. 3j). Although some still showed two bilayers, the membranes were much more separated than in *fsc1Δ* cells and the structures did not contain encapsulated cargo (Fig. 3j; quantification in Fig. 3k). Similar results were obtained after DTT treatment (Supplementary Fig. 5g–i). Taken together, these results show that Rop1 is required for proper autophagosome formation.

## Generation of high membrane curvature underlies Rop1's function in autophagy

Next we tested whether Rop1 generates high membrane curvature, like the ER-shaping REEP5/Yop1 homologs[5], despite having one less TM (Supplementary Fig. 1a). Previous experiments showed that REEP5 expressed in *E. coli* forms lipoprotein particles of extreme curvature, which do not sediment with the membrane fraction[5]. To perform similar experiments with Rop1, we used the protein from *S. japonicus* (sjRop1), as *S. pombe* Rop1 poorly expressed recombinantly. The sequences of the two proteins are highly similar. As observed with REEP5 and Yop1, a significant fraction of sjRop1 did not sediment with the membranes (Fig. 4a). Particles purified from this soluble fraction had a diameter of -14 nm when viewed by negative-stain EM (Fig. 4b, c). Although somewhat larger than particles formed by *Xenopus* REEP5 (Fig. 4b, c), their extreme curvature makes it unlikely that they are vesicles containing a lipid bilayer. Thus, like REEP5 and Yop1, sjRop1 seems to cause the conversion of a bilayer into a monolayer to form a lipoprotein particle.

Generation of high curvature was also observed when sjRop1 was purified from the membrane fraction after solubilization in detergent (Supplementary Fig. 6a) and reconstituted with phospholipids: detergent-purified sjRop1 readily formed lipoprotein particles or small vesicles (Fig. 4d, e). The APH in sjRop1 (Fig. 1a) is required for curvature generation, as is the case for REEP5 (ref. [5]), because deletions of segments of the APH (Δ107-120 or Δ103-118) resulted in large, low-curvature vesicles (Fig. 4d, e; for purity of the proteins see Supplementary Fig. 6b). Deletion of the C-terminal part of the APH (Δ120-134 or Δ126-170) had only a small effect (Fig. 4d, e). These results were confirmed by flotation experiments. When reconstituted wild-type sjRop1 was subjected to flotation in a Nycodenz gradient, most of the protein stayed near the bottom of the centrifuge tube, as expected for lipoprotein particles (Fig. 4f). In contrast, reconstituted Rop1(Δ107-120) floated to fractions expected for vesicles (Fig. 4f). We note that sjRop1 also forms homodimers just like REEP5/Yop1 (ref. [5]), as shown by photo-crosslinking with probes incorporated into the first TM

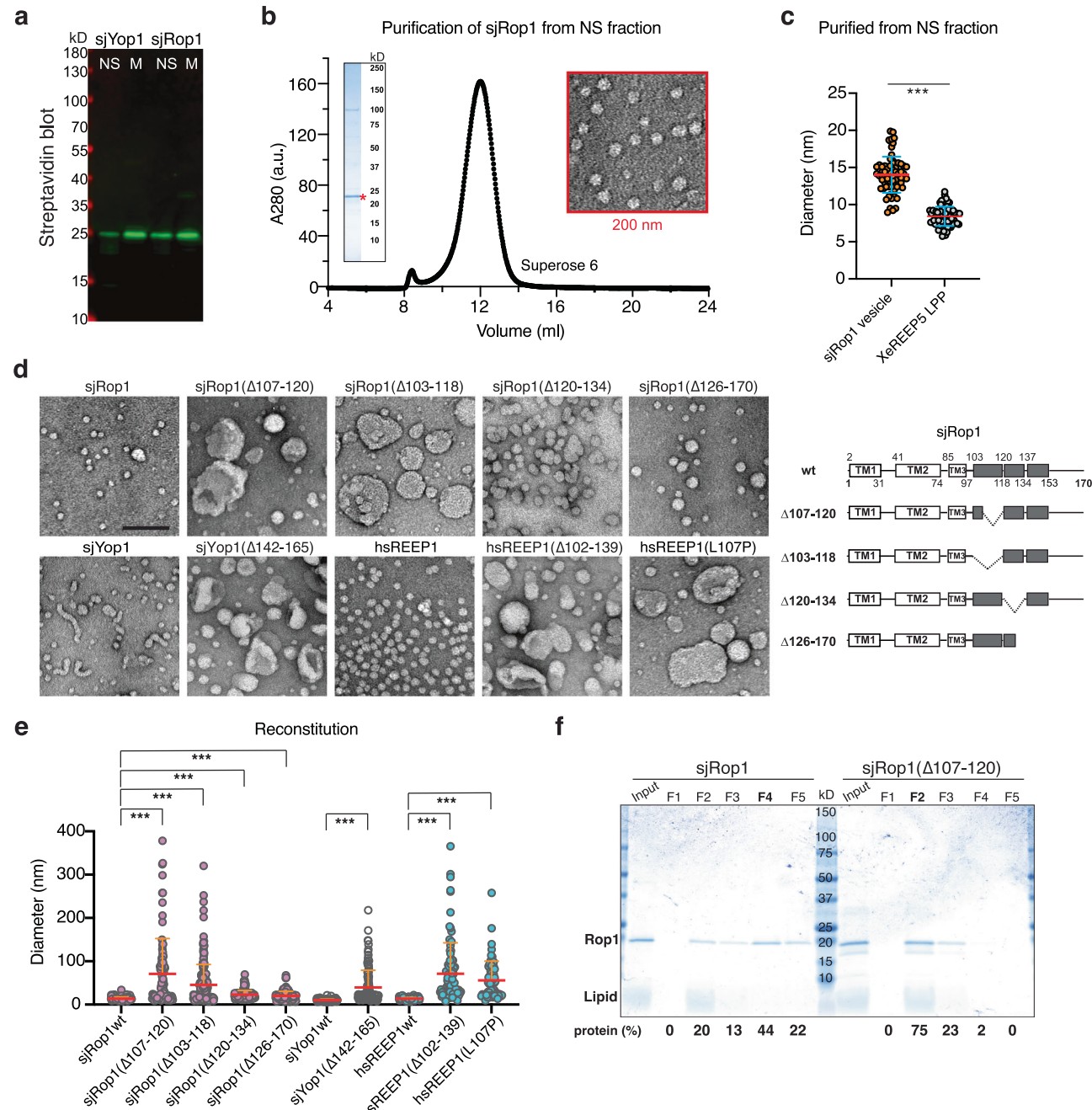

**Fig. 4 | Rop1 and its human homolog REEP1 generate high membrane curvature. a** SBP-tagged sjRop1 and sjYop1 from *S. japonicus* were expressed in *E. coli*. Cell lysates were subjected to ultracentrifugation and the membrane (M) and non-sedimentable (NS) fractions analyzed by SDS-PAGE, followed by blotting with fluorescently labeled streptavidin. **b** sjRop1 of the NS fraction was bound to streptavidin beads, the tag was removed with TEV protease, and the eluted material subjected to gel filtration. The peak fraction was analyzed by Coomassie-blue staining (left inset) and negative-stain EM (right inset). **c** Diameters of Rop1 particles compared to those of lipoprotein particles (LPP) generated in an analogous manner by *Xenopus laevis* REEP5 (XeREEP5)[5]. *** indicates a significant difference with *p*-value 2.1 × 10⁻³². calculated from a two-sample *t* test with unequal variance. **d** sjYop1, sjRop1, human REEP1 (hsREEP1) or the indicated mutants were purified from the membrane fraction and mixed with liposomes. The detergent was removed, and the reconstituted material was analyzed by negative-stain EM. Scale bar, 100 nm. The panels on the right show the domains of sjRop1. The APH is shown in gray with deleted regions indicated. Predicted breaks in the APH are indicated by vertical lines. **e** Diameters of structures seen in (**d**). For tubules, the largest width is given and for irregular structures, the largest dimension. *** indicates significant differences with *p*-values < 0.001, calculated from a two-sample *t* test with unequal variance. The exact *p*-values are listed in the Source Data file. **f** Reconstituted material as in (**d**) was subjected to flotation in a Nycodenz gradient. Fractions were analyzed by Coomassie-blue staining and protein abundance in individual fractions is expressed as percentage of total protein.

(Supplementary Fig. 6c, d). This TM is predicted to reside at the interface between the two monomers and is equivalent to the second TM of REEP5/Yop1 proteins (Supplementary Fig. 6c).

Purified human REEP1 (hsREEP1) behaved analogously to sjRop1 (Supplementary Fig. 6e). Mutations in the predicted APH (Δ102-139 or L107P) compromised REEP1's ability to generate lipoprotein particles or small liposomes (Fig. 4d, e). Notably, these mutations are associated with the human diseases distal hereditary motor neuropathy type V (HMN5B)[30] and hereditary spastic paraplegia (HSP)[31], respectively. Taken together, our data show that REEP1 subfamily members can

generate extremely high membrane curvature like the REEP5 homologs, despite the fact that they have one less TM. When present at endogenous levels in *S. pombe*, these proteins do not seem to form lipoprotein particles because both Rop1 and Yop1 were primarily found in the sedimentable membrane fraction (Supplementary Fig. 6f), as observed previously for REEP5 in *Xenopus laevis* eggs[5].

Finally, we tested whether Rop1 needs to generate high membrane curvature for its function in autophagy. Indeed, even small deletions in the APH reduced cell survival in DTT (Fig. 5a), vacuolar cleavage of Yop1-GFP (Fig. 5b, c), prevented the vacuolar accumulation of cytosolic Tdh1-mCh (Fig. 5d), and the vacuolar cleavage of the cytosolic Hsc1-GFP protein (Fig. 5e). Mutation of the AIM motif caused a moderate reduction of the vacuolar cleavage of Yop1-GFP and Hsc1-

GFP (Fig. 5b, e), likely because the mutation perturbed the C-terminal part of the APH, which makes a small contribution to curvature generation (Fig. 4d, e). Taken together, these results show Rop1's function in autophagy relies on its ability to generate high membrane curvature.

We also tested the localization of wild-type and mutant Rop1. Endogenous wild-type Rop1 fused to mNeonGreen (Rop1-mNG) localized to numerous punctae, regardless of whether autophagy was induced or not (Fig. 6a, b; Supplementary Fig. 7a, b). The bulk of these punctae were close to the cortical ER marked by Yop1, but some punctae did not colocalize with the ER marker and might be vesicles (Supplementary Fig. 7b). When overexpressed, Rop1 was no longer punctate and largely colocalized with Yop1 (Supplementary Fig. 7c), suggesting that the punctate population originates from the ER. Rop1

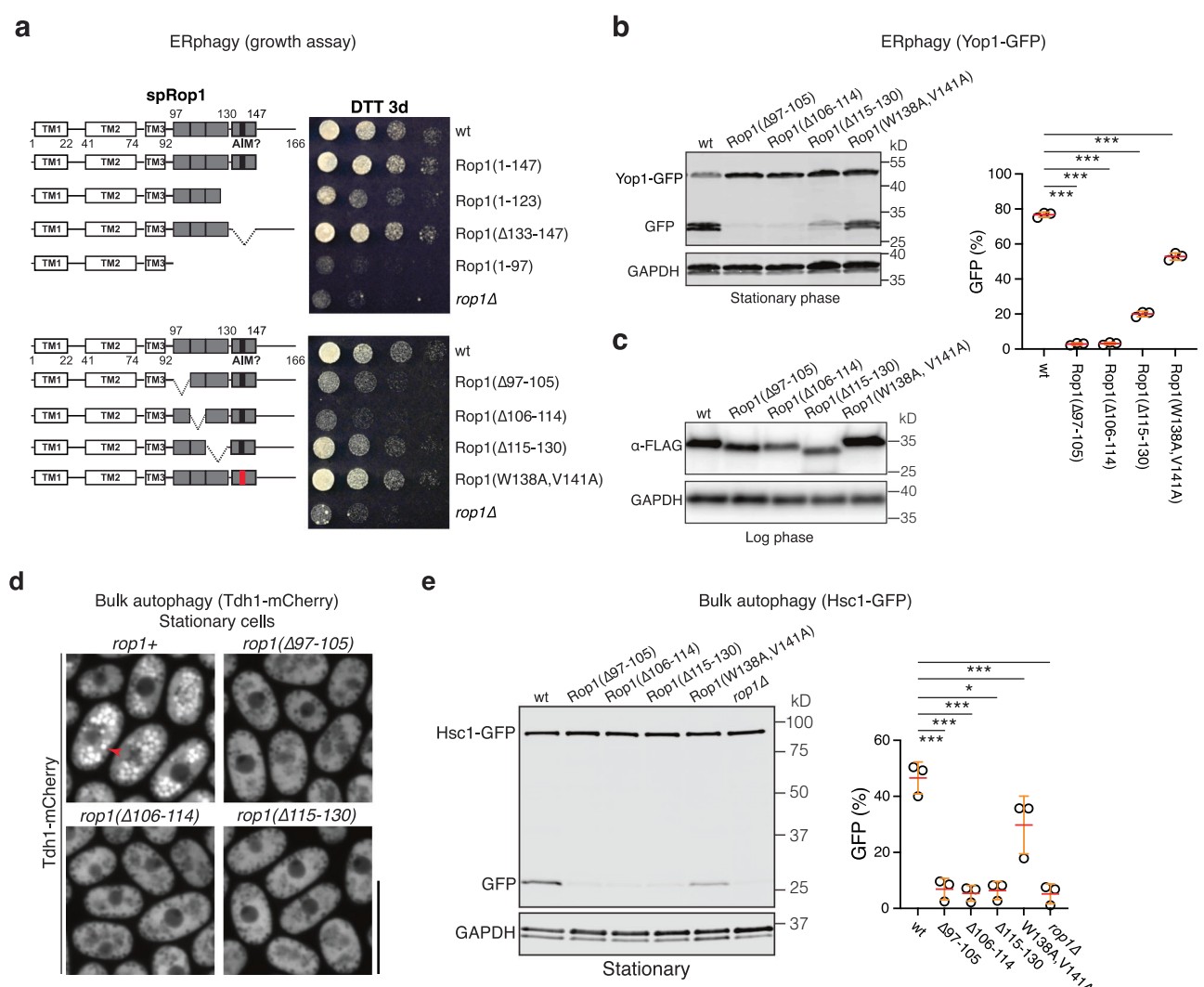

**Fig. 5 | Rop1's function in autophagy requires high membrane curvature generation. a** Wild-type (wt) *S. pombe* Rop1 (spRop1) or the indicated mutants were expressed from the endogenous genomic locus. The cells were treated with DTT for three days (DTT 3d) and plated after serial dilution. The schemes show the domains of spRop1, with the APHs and putative AIM motif in gray and black, respectively, and deleted regions indicated. The W138A, V141A mutant is designed to abrogate the AIM motif. **b** ERphagy was tested by following the cleavage of a GFP-fusion of Yop1 (Yop1-GFP) in wt cells or cells expressing the indicated Rop1 mutants. Lysates from cells at stationary phase were analyzed by SDS-PAGE and immunoblotting with GFP antibodies. Blotting with GAPDH antibodies served as loading control. The right panel shows quantification of three independent experiments (means and SD). *** indicates significant differences with *p*-values < 0.001, calculated from two-tailed Student's *t* tests. The exact *p*-

values are listed in the Source Data file. **c** wt or mutant Rop1 used in (**b**) were FLAG-tagged. Lysates from logarithmically growing cells were analyzed by immunoblotting for FLAG. **d** Bulk autophagy was tested by following the relocalization of a mCherry fusion of the cytosolic protein Tdh1 (Tdh1-mCh) expressed at endogenous levels in wt cells or cells expressing the indicated Rop1 mutants. Stationary phase cells were analyzed by fluorescence microscopy. Scale bar, 10 μm. **e** Bulk autophagy was followed by the vacuolar cleavage of a GFP-fusion of the cytosolic protein Hsc1 (Hsc1-GFP) in the indicated strains grown to stationary phase. The right panel shows the percentage of Hsc1-GFP cleavage (means and SD). **p*-value 0.07; *** significant differences with *p*-values < 0.001, calculated from two-tailed Student's *t* tests. The exact *p*-values are listed in the Source Data file.

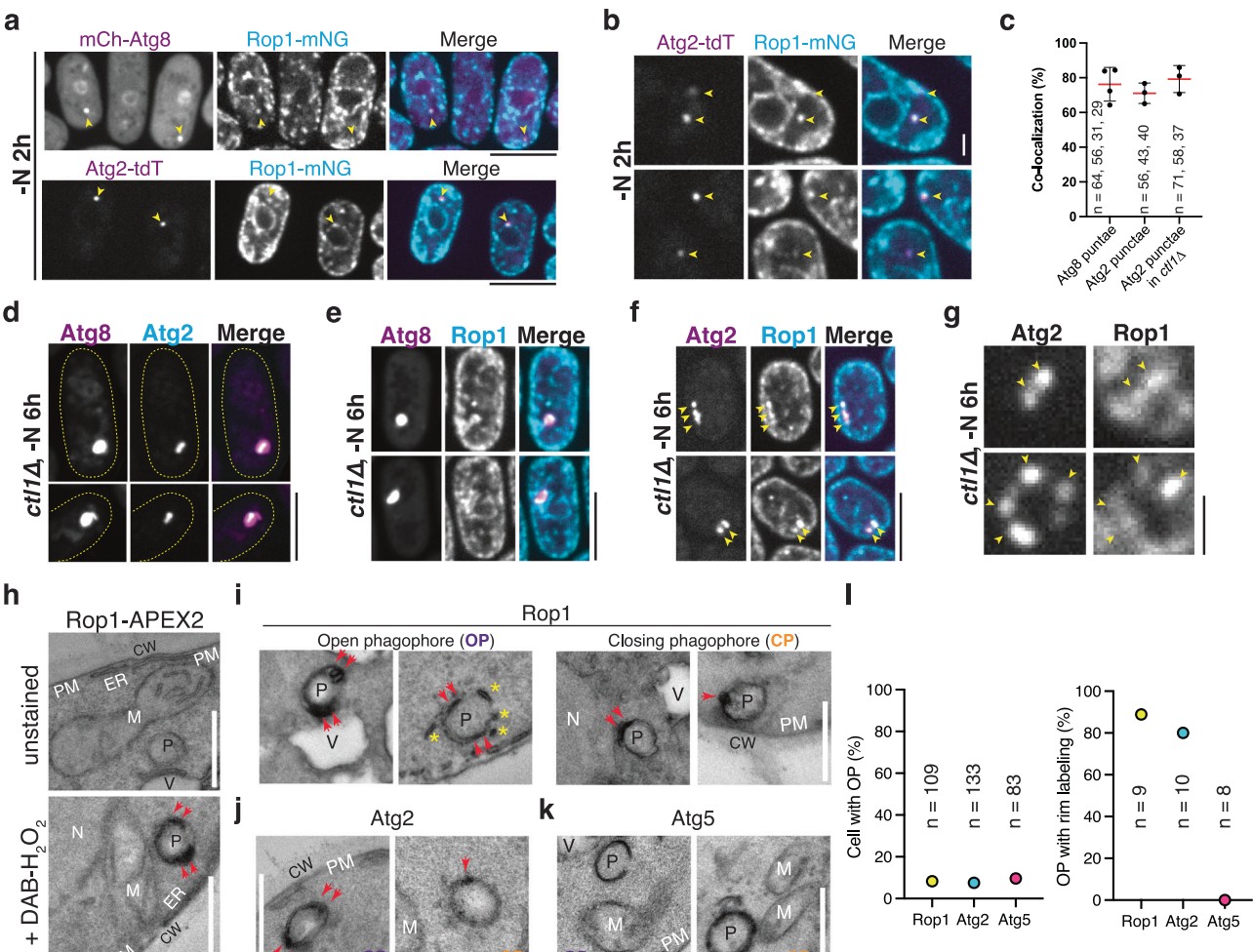

**Fig. 6 | Rop1 colocalizes with Atg2. a** A fusion of mNeonGreen with Rop1 (Rop1-mNG) was expressed at endogenous levels together with mCherry tagged Atg8 (mCh-Atg8) or tdTomato tagged Atg2 (Atg2-tdT). Cells were imaged after 2 h of nitrogen starvation (-N 2 h). Arrowheads point to punctae where colocalization is observed. Scale bar, 5 μm. **b** As in (**a**), but magnified view. Scale Bar, 1 μm. **c** Percentage of Atg2 or Atg8 punctae containing Rop1 in (**a**, **b**, and **f**). The dots show the colocalization determined in three or four experiments. *n*, number of punctae analyzed. Means and SD are shown. **d** mEGFP-Atg8 and Atg2-tdT were co-expressed at endogenous levels in *ctl1Δ* cells. Cells were imaged after 6 h of nitrogen starvation. Note that Atg2 localizes to a sub-region of the enlarged phagophores marked by Atg8 (ref. [26]). Scale bar, 5 μm. **e** As in (**d**), but for mCh-Atg8 and Rop1-mNG. **f** As in (**d**), but for Atg2-tdT and Rop1-mNG. Yellow arrowheads highlight co-

localization. **g** As in (**f**), but a magnified view. Scale bar, 1 μm. **h** Cells expressing Rop1-APEX2 were nitrogen-starved for 2 h, fixed, stained with diaminobenzidine (DAB) and H₂O₂, and analyzed by TEM. A fixed sample without DAB-H₂O₂ treatment (unstained) is shown as a control. P phagophore, CW cell wall, M mitochondrion, N nucleus, PM plasma membrane, V vacuole, ER endoplasmic reticulum. Red arrowheads point to the tips of a C-shaped phagophore. Scale bar, 500 nm. **i** As in (**h**), except that closing phagophores (CP) are shown on the right. Asterisks point to small vesicles or short tubules near the open phagophore (OP). **j** As in (**h**), but with Atg2-APEX2. **k** As in (**h**), but with Atg5-APEX2. **l** Quantification of the experiments in (**h–k**). The left panel shows the percentage of cell sections containing OPs. The right panel shows the percentage of OPs with rim labeling.

mutants carrying deletions of the APH mainly localized to punctae, some of which might again correspond to vesicles (Supplementary Fig. 7d).

### Rop1 colocalizes with Atg2 during autophagy

Following induction of autophagy by nitrogen starvation, more than 80% of phagophores marked by mCh-Atg8 or Atg2-tdT contained Rop1-mNG (Fig. 6a–c). Live-cell imaging revealed that Rop1-mNG and mCh-Atg8 are simultaneously appearing on phagophores, suggesting early Rop1 recruitment to the forming autophagosome (Supplementary Fig. 7a). These data show that a fraction of the cellular pool of Rop1 moves to phagophores after autophagy is induced.

Next, we repeated the imaging with *ctl1Δ* cells, in which phagophores are abnormally enlarged, facilitating protein localization analysis[26]. As reported before[26], mCh-Atg8 localized on bright structures that were often non-spherical, and Atg2 was seen at a subdomain of these structures (Fig. 6d). Rop1 also localized to a subregion

(Fig. 6e), and largely overlapped with Atg2 (Fig. 6f, g). Given that Atg2 localizes to the phagophore rim in yeasts[20,21,26], these results suggest that Rop1 also concentrates close to the rim.

To support this conclusion, we fused Rop1 to ascorbate peroxidase 2 (Rop1-APEX2) and examined the localization of the fusion protein by the formation of an electron-dense precipitate followed by a thin-section EM[32]. Rop1-APEX2 was visualized as dark regions at the autophagosomal membrane that were particularly intense at the tips of cup-shaped phagophores (Fig. 6h, i; quantification in Fig. 6l). Small structures near phagophores, probably Rop1-containing vesicles, were also labeled (Fig. 6i). Rop1-APEX2 labeling was also seen as one or two closely spaced dots on spherical structures that likely correspond to phagophores about to close and become autophagosomes (Fig. 6i). Atg2 showed a similarly restricted localization on these structures (Fig. 6j). In contrast, Atg5 was evenly distributed along them (Fig. 6k), consistent with the known localization of this protein over the entire phagophore[20,26]. Although

these data support the concentration of Rop1 close to the phagophore rim, the limited resolution does not exclude that Rop1 localizes to neighboring ER tubules, particularly because high-resolution EM tomography shows that highly curved ER membranes are often in immediate proximity of phagophore rims[15,16]. Consistent with this possibility, Rop1-APEX2 was also seen close to the cortical ER, and in vesicles near the ER and nuclear envelope (Supplementary Fig. 7e), in agreement with the localization of this protein by light microscopy (Supplementary Fig. 7b, d).

Finally, we tested whether Rop1's colocalization with Atg2 is required for autophagy. We asked whether Yop1, which also generates high membrane curvature but does not normally participate in autophagy, would rescue the autophagic defects in *rop1Δ* cells when bound to Atg2. To achieve association of Yop1 with Atg2, Yop1 was fused to GFP (Yop1-GFP) and Atg2 to a GFP-binding protein (GBP, a GFP nanobody)[33] (Fig. 7a). Upon starvation, a sub-population of Yop1-GFP indeed colocalized with punctae labeled with an mCherry fusion of Atg2-GBP (Atg2-GBP-mCh) (Fig. 7b). These data suggest that Yop1-GFP was either recruited to the rim of phagophores (the endogenous localization of Atg2) or was moved to an ER region that interacts with Atg2 (Fig. 7a). Importantly, relocalized Yop1 completely rescued the growth defect of *rop1Δ* cells on DTT (Fig. 7c) and increased the number of filled vacuoles in these cells to wild-type levels (Fig. 7d, e). Relocalized Yop1 also restored vacuolar cleavage of the cytosolic protein Tdh1-mCh in *rop1Δ* cells (Fig. 7f, g). The rescue required Yop1's ability to generate high membrane curvature, as relocalized Yop1-GFP lacking the APH (Δ132-189) failed to restore cell growth in the presence of DTT (Fig. 7c) and vacuolar Tdh1-mCh cleavage (Fig. 7f, g). In contrast, Yop1 with a more distal truncation (Δ143-189) that retains the APH, remained active. Both Yop1 deletion mutants relocalized to Atg2-GBP-mCh punctae in starved cells (Supplementary Fig. 7f). Rescue was not observed when Yop1-GFP was replaced with the bulk ER protein Sec63-GFP (Fig. 7c, f, g), even though it also colocalized with Atg2-GBP-mCh punctae in starved cells (Fig. 7b). These data indicate that Rop1's function in autophagy requires both its curvature-generating activity and its colocalization with Atg2.

Similar results were obtained when Yop1-GFP was combined with a GBP-fusion of Atg18b (Atg18b-GBP)[26], a known interaction partner of Atg2. Although not as pronounced, rescue of the autophagy defect of *rop1Δ* cells was again observed (Fig. 7c–g). On the other hand, the homolog Atg18a was ineffective (Fig. 7c; Supplementary Fig. 7f), consistent with reports that it differs from Atg18b in localization and function[26]. Taken together, these data show that Rop1 needs to colocalize with Atg2/Atg18b during autophagy. Like Atg2, Rop1 likely functions close to the highly-curved phagophore rim.

To test whether Rop1 physically interacts with Atg2 or other known autophagy factors, we performed pull-down experiments using cells expressing FLAG-tagged Rop1 (Rop1-FLAG) at endogenous levels and identified interacting proteins by mass spectrometry (Supplementary Fig. 8). The cells were grown either to logarithmic phase or starved, and controls were performed with Yop1-FLAG under identical conditions. These experiments showed that Rop1-FLAG did not pull down Atg2 or other autophagy components; a few peptides of Atg9 were detected, but probably arise from non-specific interactions, as they were seen with both Rop1-FLAG and Yop1-FLAG. Thus, Rop1 does not interact with the cytosolic autophagy machinery. Taken together, our results indicate that colocalization, but not physical association, of Rop1 with Atg2 is required for autophagy.

## Discussion
Here, we demonstrate that Rop1, a member of the conserved REEP1 protein family, is involved in all forms of autophagy in *S. pombe*, including autophagy of the ER, mitochondria, peroxisomes, and cytosolic proteins. Rop1 is crucial for autophagosome formation, recruited at early stages to a subdomain of phagophores, and required

for their maturation into autophagosomes. Rop1 function relies on its ability to generate high membrane curvature and on its colocalization with Atg2. Based on these results, we propose that Rop1 facilitates the formation and growth of the double-membrane structure of the autophagosome. Our results show that Rop1 is not a functional homolog of Atg40 in *S. cerevisiae*, which has a more limited role as an ERphagy receptor. Rop1 does not have primary sequence orthologs in budding yeast, but it is possible that it is functionally replaced by other proteins.

Our experiments indicate that Rop1's function in autophagy requires its colocalization with Atg2. Atg2 has been reported to localize to the rim of growing phagophores in different yeasts[20,21,26], but the best evidence comes from imaging experiments in *S. cerevisiae*, where unusually large phagophores are generated in cells overexpressing components of the Cvt pathway[20]. This pathway does not seem to exist in *S. pombe*, but large phagophores are formed in *ctl1Δ* cells, where Atg2 localizes to a subdomain[26] that likely is the rim. However, Atg2's localization in wild-type cells is less clear. Furthermore, it is not understood how Atg2 is recruited from the cytosol to the phagophore rim. Given these uncertainties and the limited resolution of our own imaging experiments, the localization of Rop1 also requires further study.

One possibility is that Rop1 localizes to the rim of phagophores and stabilizes its high membrane curvature (Fig. 7h). The identification of phagophore rim-stabilizing proteins has been a long-standing question[19], and Rop1 would be a good candidate, as it is the only integral membrane protein involved in autophagy that belongs to a family of known curvature-stabilizing proteins. Moreover, Rop1 homologs are found in most eukaryotic cells. Our observation that autophagy can be restored in a *rop1Δ* mutant by forcing Yop1 to associate with Atg2 (Fig. 7a–g) could be interpreted by Yop1 being dragged from the ER to the phagophore rim, where it would replace Rop1 in stabilizing the high membrane curvature. Because there is no obvious membrane continuity between the ER and phagophores[15,16], one would have to assume that Yop1 is moved in vesicles, possibly by COPII-mediated transport[34]. This model would explain why dragging Sec63 to the rim does not restore autophagy in the *rop1Δ* strain because Sec63 cannot stabilize high membrane curvature. It is unlikely that stabilization of the rim curvature is the only mechanism that keeps the two membranes of a phagophore in close proximity, because the intermembrane distance is smaller in the phagophore body than at the rim and decreases even further as the phagophore grows[15,16], perhaps by the influx of lipids but not luminal content. Nevertheless, our experiments show that autophagosomes with closely apposed membranes cannot form without Rop1. Thus, keeping the two membranes in close proximity may be required to initiate the formation of proper phagophores. In the absence of Rop1, some cargo might occasionally be encapsulated in aberrant autophagosomes, which could explain why some forms of autophagy are not entirely blocked.

An alternative possibility is that Rop1 localizes to specialized, high curvature regions of the ER which abut the rim of phagophores. Close proximity between these high-curvature membranes is indicated by EM tomography data in both *S. cerevisiae* and mammalian cells[15,16]. In this model, Rop1 would stabilize the curvature of the tips of ER tubules or of other high-curvature regions of the ER. Such ER regions could facilitate the Atg2-mediated lipid flow from the ER into phagophores[22,23] (Fig. 7h). Atg2 probably receives its lipids from lipid scramblases in the ER[35], which are distributed throughout the entire ER[36], but Atg2 might still prefer to channel lipids from highly curved ER regions to highly curved phagophore rims. Our rescue of the *rop1Δ* mutant would be explained by assuming that Yop1 is dragged by Atg2 from the shaft of ER tubules to their tips or other specialized regions, where it would stabilize their curvature. Again, dragging Sec63 instead would not work because it cannot generate high curvature. Although both Yop1 and Rop1 stabilize high membrane

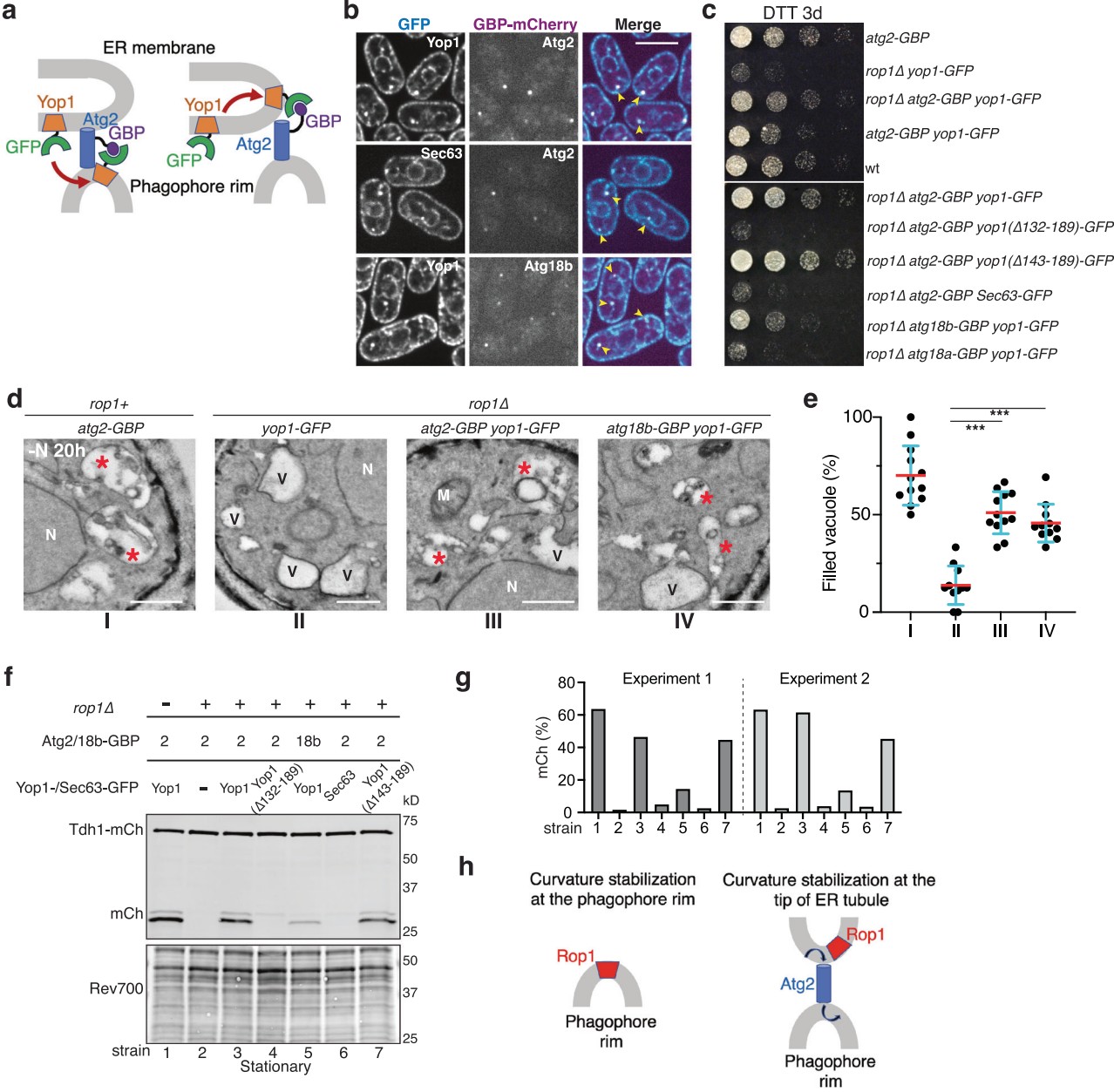

**Fig. 7 | Colocalization of Rop1 with Atg2 is required for autophagy. a** Scheme showing how a fusion of Atg2 to a GFP nanobody (Atg2-GBP) drags Yop1-GFP from the ER to the phagophore rim or to the tips of ER tubules. A similar strategy was used with Atg18b-GBP. **b** Atg2-GBP or Atg18b-GBP were tagged with mCherry and expressed together with Yop1-GFP or Sec63-GFP. The cells were imaged after 2 h of nitrogen starvation (-N 2 h). Arrowheads point to colocalization of the coexpressed proteins. Scale bar, 5 μm. **c** Wild-type (wt) *S. pombe* cells or cells expressing the indicated mutants at endogenous levels were treated with DTT for three days (DTT 3d) and plated after serial dilution. **d** wt or *rop1Δ* cells expressing the indicated fusion proteins were nitrogen-starved for 20 h and analyzed by TEM. Asterisks point to vacuoles containing electron-dense material, likely residual membranes of autophagosomes. V empty vacuole, M mitochondrion, N nucleus. Scale bars, 500 nm. **e** Quantification of the results in (**d**). Shown is the percentage of filled

vacuoles per cell as individual measurements, means, and SD. *** indicates significant differences with *p*-values < 0.001, calculated from two-tailed Student's *t* tests. The exact *p*-values are listed in the Source Data file. **f** Bulk autophagy was tested by vacuolar cleavage of a mCherry fusion of the cytosolic protein Tdh1 (Tdh1-mCh) in cells expressing Atg2-GBP or Atg18b-GBP together with GFP fusions of full-length Yop1, Yop1 carrying a deletion of the APH (Δ132-189), Yop1 carrying a deletion that retains the APH (Δ143-189), or Sec63. Lysates of cells grown to stationary phase were analyzed by blotting for mCherry. The lower panel shows a loading control in which proteins were stained with Revert 700 (Rev700).
**g** Quantification of two experiments as in (**f**). **h** Two possible models for the function of Rop1 in autophagosome formation. Rop1 could stabilize the high membrane curvature of the phagophore rim or that of a specialized ER region and facilitate Atg2-mediated lipid flow from the ER to the rim of phagophores (arrows).

curvature, they probably reside in distinct ER regions, because normally Yop1 plays no role in autophagy. The localization of Rop1 to the tips of ER tubules might also facilitate the shedding of small vesicles, which in turn could serve as substrates for ERphagy. A dual role of Rop1 in facilitating lipid channeling and generating vesicles might explain why the absence of Rop1 affects ERphagy more than other autophagy pathways. It should be noted that in both models

depicted in Fig. 7h the colocalization of Rop1 and Atg2 would be caused indirectly, as the two proteins do not physically interact with one another (Supplementary Fig. 8). Perhaps, the high membrane curvature generated by Rop1 can concentrate Atg2.

During the revision of this manuscript, a preprint by Zou et al.[37] also reported that Rop1 (called Yep1 in that study) is required for ERphagy. Both studies agree that Rop1's function in autophagy

requires high membrane curvature generation. However, we found that all forms of autophagy are affected by the absence of Rop1, demonstrated with multiple substrates expressed at endogenous levels and by quantitative proteomics experiments. In contrast, Zou et al. found only a moderate effect on bulk autophagy with two tested substrates. The reason for the discrepancy is unclear. In addition, we observed that *S. cerevisiae* Atg40, expressed from the endogenous Rop1 promoter, cannot substitute for Rop1 in *S. pombe*, whereas Zou et al. reported complementation when Atg40 was expressed at elevated levels. Both studies agree that Rop1 differs from Atg40 by not being an ERphagy receptor.

It is possible that the mammalian REEP1-4 proteins in higher organisms have a similar function as Rop1 in fission yeast, but this requires further study. The mammalian REEP1 proteins have been reported to interact with microtubules and facilitate lipid droplet formation[6,38,39], and how these functions relate to a possible role in autophagy remains to be investigated. Nevertheless, our results raise the possibility that autophagy defects may underly the disease pathologies of HSP and the related disorder HMN5B caused by mutations in REEP1 or REEP2[30,31,40]. Several REEP1 disease mutations map to its curvature-generating APH and thus might compromise autophagy.

## Methods

### Strain construction and autophagy analysis

Supplementary Table 1 lists the *S. pombe* strains used in this study. Standard genetic and PCR-based gene targeting methods, using pFA6a integration and tagging cassettes, were used to transform yeast cells and to construct strains[41]. All knock-out strains have the coding sequence completely removed from their genomic locus. An antibiotics marker was inserted into the coding region, as confirmed by PCR. C-terminally tagged Yop1, Rtn1, Tom20, Pex11, Sfb3, Sec24, Tdh1, Hsc1, Pyk1, Pgk1, and Rop1 were expressed under the control of their endogenous promoters from their chromosomal loci. N-terminally tagged Atg8 was expressed either under the control of the inducible nmt1 or 41nmt1 promoters or expressed from its native promoter, all from the native chromosomal locus. To perform time-lapse imaging of Rop1-mNG and mCherry-Atg8, C-terminally tagged Rop1 was expressed from the inducible 81nmt1 promoter to achieve moderate overexpression. To express YFP, EYFP under the control of the nmt1 promoter was inserted into pJK148 (Supplementary Table 2) and integrated into the *leu1* locus. Double or triple-mutant strains were generated by genetic crossing and random sporulation. Because of lower mating efficiency, double and triple mutants containing *fsc1Δ* were constructed by two rounds of transformation. *S. japonicus* strains were generated as previously described[5].

To test for autophagy defects by microscopy analysis or proteolytic cleavage assays, *S. pombe* colonies from a freshly streaked plate were grown overnight in liquid Edinburgh minimal medium (EMM5S with 2% glucose, Sunrise Science) at 30 °C. The next day (day 1), cultures were diluted to 0.2 OD600 both early AM and late PM to generate cells in logarithmic growth phase. On the next day, the cultures were diluted to OD600 of 0.5 and used for analysis. For stationary phase analysis, cells were grown without dilution for another 32–36 h at 30 °C. For DTT treatment, a final concentration of 10 mM DTT was added to the diluted cells and the cells were grown at 30 °C for the indicated time periods. For nitrogen-starvation, cells corresponding to ~3 OD600 units were washed once with EMM-N medium with 2% glucose (Sunrise Science), resuspended in EMM-N medium at OD600 of 0.5, and grown at 30 °C for the indicated time periods. YE5S (yeast extract with five supplements) was used for *S. japonicus* cell cultures. The cells were grown in YE5S for ~24 h at 30 °C with proper dilutions to keep the culture in log phase. 200 ng/μl of rapamycin was added and the cells were cultured for another 20 h before imaging.

For imaging, cells corresponding to 1–2 OD600 units were sedimented at $4000 \times g$ for 30 s and the supernatant was removed, leaving ~100 μl. The cells were then resuspended for fluorescence imaging. Imaging was carried out with a spinning disk mounted on a Ti-motorized inverted confocal microscope (Nikon), as previously described[5]. Images were acquired with a cooled CCD camera (Hamamatsu Photonics) controlled with MetaMorph (version 7.8) or NIS-Elements. All display and analysis functions were performed with ImageJ. To quantify colocalization of Rop1 and Atg proteins, a 5 × 5 pixel area (~0.32 × 0.32 μm) centered at the brightest Atg pixel was drawn on both channels; the complete or partial overlay of the two squares was binned as colocalization.

To assay for fluorescent protein cleavage, cells corresponding to 3–6 OD600 units were pelleted and washed once with cold water containing 1 mM phenylmethylsulfonyl fluoride (PMSF). The cells were resuspended in 500 μl of cold water and left on ice for 5 min. Then, 75 μl of 1.85 M NaOH, 7.5% (v/v) 2-mercaptoethanol were added to lyse the cells. Next, 75 μl of 55% trichloroacetic acid was added to precipitate the proteins. After centrifugation at $16,000 \times g$ for 10 min at 4 °C, the supernatant was removed, and the pellet was dried at room temperature for 15 min. The pellet was then suspended in 75 μl of sample buffer containing 8 M urea, 200 mM Tris-HCl pH 6.8, 1 mM EDTA, 5% (w/v) SDS, 0.1% (w/v) bromophenol blue, 1.5% (w/v) DTT. The samples were subjected to 4–20% SDS-PAGE and analyzed by immunoblotting (IB) with the following primary antibodies: anti-GFP mouse monoclonal antibody (JL-8, Takara, 1:2000 for IB), GFP rabbit polyclonal antibody (PABG1, ChromoTek, 1:4000 for IB) for mEGFP detection, anti-mCherry mouse monoclonal antibody (1C51, Abcam, 1:2000 for IB), and anti-GAPDH mouse monoclonal antibody (GA1R, Abcam, 1:10000 for IB). Secondary antibodies were used at 1:5,000 or 1:10,000 dilution. Western blots were imaged using an Amersham Imager 600 with the ImageQuant software or using a LI-COR Odyssey CLx imager with the LI-COR Acquisition software. Staining for total protein was carried out with Revert™ 700 total protein stain kit (LI-COR) according to the instructions of the manufacturer.

DTT-induced ER-stress sensitivity assays were performed as previously described[24]. Cells were grown in liquid EMM5S + 2% glucose at 30 °C with 10 mM DTT for 3 days. Cells were washed once with fresh EMM5S before plating. Five-fold dilutions of the indicated strains were spotted onto YE5S plates. Plates were incubated at 30 °C and imaged after colony formation.

### Quantitative isobaric tag-based proteomics

*S. pombe* cells were nitrogen-starved for 16 h and ~15 OD600 units of cells were pelleted and frozen in liquid N₂. Three biological replicates for each strain were collected on separate days, and all replicates were processed on the same day. Frozen cell pellets were washed with water and resuspended in 1 ml of lysis buffer containing 8 M urea, 200 mM EPPS pH 8.5 and protease inhibitors (Pierce A32953). Cells were then lysed by bead-beating and the lysates were clarified by centrifugation at $2000 \times g$ for 30 s. The total protein concentration was determined using the Pierce BCA Protein (23227) assay. The sample was reduced with 5 mM tris(2-carboxyethyl) phosphine (TCEP) for 30 min, alkylated with 10 mM iodoacetamide for 30 min, and quenched with 10 mM DTT for 15 min. Approximately 100 μg of protein were transferred to a new tube for methanol-chloroform precipitation. The pellet was resuspended in 200 mM EPPS, pH 8.5 and digested at room temperature for 14 h with LysC protease at a 100:1 protein-to-protease ratio. Trypsin was then added at a 100:1 protein-to-protease ratio and the reaction was incubated for 6 h at 37 °C. Streamlined–TMT labeling and LS-MS were all done following the protocol described in refs. 42,43. Briefly, TMTpro reagents (0.8 mg) were dissolved in anhydrous acetonitrile (40 μL) of which 7 μL was added to the peptides (50 μg) with 13 μL of acetonitrile to achieve a final concentration of approximately 30% (v/v). Following incubation at room temperature for 1 h, the reaction was quenched with hydroxylamine at a final concentration of 0.3% (v/v). TMTpro-

labeled samples were pooled at a 1:1 ratio across all 15 samples. For each experiment, the pooled sample was vacuum centrifuged to near dryness and subjected to C18 solid-phase extraction (SPE) (Sep-Pak, Waters).

The pooled, labeled peptide sample was subjected to basic pH reversed-phase fractionation[44] using an Agilent 300Extend C18 column (3.5 µm particles, 4.6 mm ID, and 220 mm in length) and Agilent 1200 pump equipped with a degasser and UV detector. Peptides were eluted with a 50-min linear gradient from 5% to 35% acetonitrile in 10 mM ammonium bicarbonate pH 8 at a flow rate of 0.6 mL/min and fractions were further processed by acidification with 1% formic acid and vacuum centrifugation to near dryness[45]. Each fraction was desalted via StageTip, dried again via vacuum centrifugation, and disolved in 5% acetonitrile, 5% formic acid for LC-MS/MS analysis.

Mass spectrometric data were collected on an Orbitrap Eclipse mass spectrometer coupled to a Proxeon NanoLC-1200 UHPLC. The 100 µm capillary column was packed with 35 cm of Accucore 150 resin (2.6 µm, 150 Å; ThermoFisher Scientific). The scan sequence began with an MS1 spectrum (Orbitrap analysis, resolution 60,000, 350−1350 Th, automatic gain control (AGC) target $4 \times 10^5$, maximum injection time 100 ms). Data were acquired ~90 min per fraction. MS2 analysis consisted of collision-induced dissociation (CID), quadrupole ion trap analysis, automatic gain control (AGC) $2 \times 10^4$, NCE (normalized collision energy) 35, q-value 0.25, maximum injection time 120 ms), isolation window at 0.5 Th, and TopSpeed set to 3 sec. RTS was enabled and quantitative SPS-MS3 scans (resolution of 50,000; AGC target $2.5 \times 10^5$; collision energy HCD at 55%, max injection time of 250 ms) were processed through Orbiter with a real-time false discovery rate filter implementing a modified linear discriminant analysis using the *S. pombe* database. For FAIMS, the dispersion voltage (DV) was set at 5,000 V, the compensation voltages (CVs) used were −40 V, −60 V, and −80 V, and the TopSpeed parameter was set at 1 s.

Spectra were converted to mzXML via MSconvert (Comet 2021.01.0)[46]. Database searching included all entries from the *S. pombe* UniProt reference Database (https://www.uniprot.org/proteomes/UP000002485; downloaded: August 2021). The database was concatenated with one composed of all protein sequences for that database in reversed order. Searches were performed using a 50-ppm precursor ion tolerance for total protein level profiling. The product ion tolerance was set to 0.9 Da. The wide mass tolerance windows were chosen to maximize sensitivity in conjunction with Comet searches and linear discriminant analysis[47,48]. TMTpro labels on lysine residues and peptide N-termini (+304.207 Da), as well as carbamidomethylation of cysteine residues (+57.021 Da) were set as static modifications, while oxidation of methionine residues (+15.995 Da) was set as a variable modification. Peptide-spectrum matches (PSMs) were adjusted to a 1% false discovery rate (FDR)[49,50]. PSM filtering was performed using a linear discriminant analysis, as described previously[48] and then assembled further to a final protein-level FDR of 1%[50]. Proteins were quantified by summing reporter ion counts across all matching PSMs[51]. Reporter ion intensities were adjusted to correct for the isotopic impurities of the different TMTpro reagents according to manufacturer specifications. The signal-to-noise measurements of peptides assigned to each protein were summed and these values were normalized so that the sum of the signal for all proteins in each channel was equivalent. Each protein abundance measurement was scaled, such that the summed signal-to-noise for that protein across all channels equals 100, thereby generating a relative abundance measurement (see Source Data file). In all experiments, the total number of protein species detected are the same (>4000). Data analysis was done by Graphpad Prism and the volcano plots were generated at https://huygens.science.uva.nl/VolcaNoseR/.

### Label-free mass spectrometry analysis
To identify binding partners of Rop1, FLAG-tagged Rop1 or Yop1 were expressed at endogenous levels in cells maintained in YES rich medium or in EMM-N medium for 2 h. ~500 OD of cells were collected and washed once with water. Cells were lysed with Freezer/mill in lysis buffer containing 25 mM Tris pH 8.0, 150 mM NaCl, and protease inhibitors. The membrane fraction was isolated from the cell lysates and solubilized in lysis buffer containing additional 1% N-dodecyl-β-maltoside (DDM) for 1 h at 4 °C. ~100 µl of M2 resin was then incubated with DDM-solubilized supernatant for 1 h at 4 °C. The resin was washed with 10 volumes of lysis buffer containing 0.05% DDM and the proteins were eluted with 3 volumes of elution buffer containing 25 mM Tris pH 8.0, 150 mM NaCl, 0.03% DDM, 0.4 mg/ml 3xFLAG peptide. 1/100 of eluted samples was subject to SDS-PAGE analysis to estimate the protein concentration. ~50 ng of protein was TCA precipitated, the pellet was washed once with acetone and once with methanol.

All samples were resuspended in 100 µL of 100 mM HEPES, pH 8.5 and digested at 37 °C with trypsin at a 100:1 protein-to-protease ratio overnight. The samples were desalted via StageTip, dried by vacuum centrifugation, and dissolved in 5% acetonitrile, 5% formic acid for LC-MS/MS processing.

Mass spectrometry data were collected using a Exploris 480 mass spectrometer (Thermo Fisher Scientific, San Jose, CA) coupled with a Proxeon 1200 Liquid Chromatograph (Thermo Fisher Scientific). Peptides were separated on a 100 µm inner diameter microcapillary column packed with ~25 cm of Accucore C18 resin (2.6 µm, 150 Å, Thermo Fisher Scientific). ~1 µg was loaded onto the column. Peptides were separated using a 90 min gradient of 4 to 30% acetonitrile in 0.125% formic acid with a flow rate of 520 nL/min. The scan sequence began with an Orbitrap MS1 spectrum with the following parameters: resolution 60,000, scan range 350−1350 Th, automatic gain control (AGC) target "standard", maximum injection time "auto", RF lens setting 40%, and centroid spectrum data type. The top twenty precursors were selected for MS2 analysis which consisted of HCD high-energy collision dissociation with the following parameters: resolution 15,000, AGC was set at "standard", maximum injection time "auto", isolation window 0.7 Th, normalized collision energy (NCE) 28, and centroid spectrum data type. In addition, unassigned and singly charged species were excluded from MS2 analysis and dynamic exclusion was set to 90 s.

Mass spectra were processed using a Comet-based in-house software pipeline. MS spectra were converted to mzXML using a modified version of ReAdW.exe. Database searching with *S. pombe* UniProt database was performed as described above. PSM filtering was performed using a linear discriminant analysis, as described previously[48], while considering the following parameters: XCorr, ΔCn, missed cleavages, peptide length, charge state, and precursor mass accuracy. Peptide-spectral matches were identified, quantified, and collapsed to a 1% FDR and then further collapsed to a final protein-level FDR of 1%. Protein assembly was also guided by principles of parsimony to produce the smallest set of proteins necessary to account for all observed peptides. Spectral counts were then extracted and the data were subsequently analyzed by Venny 2.1.0 (https://bioinfogp.cnb.csic.es/tools/venny/).

### Thin-section transmission electron microscopy (TEM)
EM of fission yeast was performed as previously described with some modifications[52]. Cells corresponding to approximately 10 OD600 units were mixed with 1/25 volume of 25% glutaraldehyde and immediately sedimented at ~1500 × *g* for 5 min. The cells were washed once with water and resuspended in freshly prepared 4% KMnO₄. After incubation at 4 °C for 3 h, the cells were pelleted, washed extensively with water, and resuspended in 2% uranyl acetate and left rotating at 4 °C for 16 h. The cells were again washed extensively with water and dehydrated through a graded ethanol series and embedded in Spurr's resin. Thin sections were examined using the Tecnai G² Spirit BioTWIN operated at an acceleration voltage of 80 kV or JEOL 1200EX. Images were acquired with AMT imaging software and processed with ImageJ.

For quantification of filled vacuole (%), vacuole #, and autophagic structure #/cell, cells with similar lengths and a clear nucleus visualized in the center were chosen to minimize natural variations in different cross-sections of the cell. Circularity of individual autophagic structures was measured by ImageJ. Data were plotted using Graphpad Prism.

Sample preparation for diaminobenzidine (DAB) staining EM was performed as in ref. 53 with some modifications. ~15 OD of starved *S. pombe* cells were harvested and washed once with cold water. The cell pellets were spheroplasted by resuspending in 1 ml of 1.2 M sorbitol containing 10 mg/ml of Zymolyase-20T and rotated at room temperature for 10 min. Cells (1 ml) were then layered on top of a 200 µl cushion of 4% PFA /0.2% glutaraldehyde (in 0.1 M sodium phosphate buffer, pH 7.4) and immediately pelleted. Next, the supernatant was removed and fixed in 4% PFA/0.2% glutaraldehyde for 1 h and then washed with PBS. Half of the fixed spheroplasts was stained for 30 min in a 0.05% DAB-HCl solution with 1:1,000 of 30% $H_2O_2$ to obtain an electron-dense staining caused by the APEX2 tag; the other half was used as a non-treated control. The stained sample was washed in PBS, then incubated in filtered 6% potassium permanganate for 1 h. Pellets were washed in PBS until the supernatant was clear.

Samples were dehydrated using solutions of increasing ethanol (EtOH) concentrations for 15 min each (7%, 30%, 50%, and 70% [once each] and 100% EtOH [twice]). After the 50% EtOH step, samples were additionally incubated in 1% $OsO_4$ /50% EtOH for 1 h. Subsequent resin embedding was done using low-viscosity Spurr's solution.

### Protein purification and in vitro reconstitution

*S. japonicus* Yop1, Yop1(Δ142-165), Rop1, Rop1(Δ107-120), Rop1 (Δ126-170), human REEP1, REEP1(Δ102-139), and REEP1(L107P) were expressed as SBP fusion proteins in *E. coli* BL21-CodonPlus (DE3)-RIPL cells (Agilent) and purified as previously described for Yop1/REEP5 proteins[5]. All generated plasmids used in this study are listed in Supplementary Table 2. *S. japonicus* Rop1 (sjRop1) lipoprotein particles (LPPs) were purified from the non-sedimentable fraction after 1 h of ultracentrifugation in a Ti-45 rotor (Beckman) at ~230,000 x g at 4 °C, as described[5]. The membrane fractions containing the REEP1 proteins were solubilized in DDM and the proteins were purified, as described[5]. sjRop1 LPPs and the purified REEP1 proteins from the membrane fractions were further purified by size exclusion chromatography (SEC) on Superose 6 or Superdex 200 (GE Healthcare) columns, respectively.

To incorporate DDM-solubilized proteins into liposomes, protein-free liposomes were first generated with an *S. cerevisiae* Yeast Polar Lipid extract (Avanti) and protein was then added at a 1:300 (protein:lipid) molar ratio with DDM supplemented to a final concentration of ~0.1%. The mixture was incubated with gentle rotation at 4 °C for 1 h. The detergent was then removed by four successive additions of Bio-Beads SM-2 resin (Bio-Rad) over the course of ~24 h at 4 °C. Insoluble aggregates were removed by centrifugation.

Reconstituted wild-type or mutant Rop1 was subjected to flotation in a discontinuous Nycodenz gradient (40, 30, 20, 10, and 0%) as described previously[5]. Fractions were collected from the top and analyzed by SDS-PAGE and Coomassie-blue staining.

### Negative-stain EM

Five µl of reconstituted proteoliposomes or purified sjRop1 LPPs were added to a glow-discharged carbon-coated copper grid (Pelco, Ted Pella Inc.) for 1 min. Excess sample was blotted off with filter paper. The grids were washed once with deionized water, and then stained once with freshly prepared 1% uranyl acetate for 30 s prior to blotting and air-drying. Images were collected using a Tecnai G² Spirit BioTWIN transmission electron microscope operated at an acceleration voltage of 80 kV, or a JEOL 1200EX microscope equipped with an AMT 2k CCD camera.

### Site-specific photo-crosslinking

To incorporate photoreactive Bpa probes into sjRop1, amber codons were introduced into Rop1 at various positions (see Supplementary Table 2 for plasmids). The mutated Rop1 expression plasmid and the amber suppressor tRNA plasmid were co-transformed into BL21 (DE3) cells (NEB), as described[5]. One mM final concentration of H-Bpa-OH (Chempep) was added to the medium, and the amber suppressor tRNA was induced 1–2 h before induction of sjRop1 expression.

For photo-crosslinking, 20 µl of a membrane suspension in 25 mM HEPES/KOH pH 7.5, 150 mM NaCl was added to Thermowell PCR tubes (Corning) and placed onto an ice-cold metal block. The samples were irradiated for 30 min with a long-wave UV lamp (Blak-Ray) and then analyzed by SDS-PAGE and Western blotting with DyLight 800-labeled streptavidin (Invitrogen) diluted to 1:4000 in 1% BSA.

### Pull-down experiments with purified Rop1 and Atg8

To test for interactions between Rop1 and Atg8 family proteins, *S. cerevisiae* Atg8 was tagged at the N-terminus with His10 and *S. pombe* Atg8 was tagged at the N-terminus with GST (see Supplementary Table 2). The proteins were expressed in *E. coli* and purified on Ni-NTA and GSH beads, respectively, essentially as described[54]. The proteins were further purified by SEC on a Superdex 75 (GE Healthcare) column. SBP-tagged full-length sjRop1, scAtg40, or mutants with mutated putative AIM motifs [sjRop1(AIM) and scAtg40(AIM)] were expressed in *E. coli*. The DDM-solubilized membrane fractions were incubated with streptavidin beads and washed extensively with wash buffer containing 25 mM HEPES/KOH pH 7.4, 150 mM NaCl, 0.05% DDM, and protease inhibitors. The beads were then incubated with purified Atg8 in buffer containing 0.05% DDM for 1 h at 4 °C. The beads were washed again with 20 column volumes of buffer and eluted with one volume of 2× Laemmli sample buffer (Bio-Rad). Thirty % of the eluted sample was subjected to SDS-PAGE gel and Coomassie blue staining. A sample of the input material (5%) was analyzed in parallel.

### Statistics and reproducibility

All experiments that show statistics were independently repeated at least three times. The individual data are shown together with means and SD. The significance analysis between different groups was determined either by two-tailed Student's $t$ tests or by a two-sample $t$ tests with unequal variance, as detailed in the figure legends. $p$-values are given in the legends or in the Source Data file. Where no statistics is shown, images were obtained in at least two independent experiments and representative results are shown. The statistical analysis of the proteomics data is described in the Methods.

### Reporting summary

Further information on research design is available in the Nature Portfolio Reporting Summary linked to this article.

## Data availability

The mass spectrometry proteomics data have been deposited to the ProteomeXchange Consortium via the PRIDE partner repository with the dataset identifier PXD043260. Images are available from the corresponding author upon request. Source data are provided with this paper.

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

## Acknowledgements

The authors thank J. Waters and the Nikon Imaging Center, and M. Ericsson and the Electron Microscopy Facility, both at Harvard Medical School, for technical assistance and use of microscopes. We thank D. Moazed, J.Q. Wu, and S. Oliferenko for strains and plasmids, and D. Pellman, W. Harper, D. Zhao, M. Bao, M. Skowyra, and Z. Ji for critical reading of the manuscript. This work was supported by an NIH/NIGMS grant (R01 GM132129) to J.A.P. and an NIGMS grant (R01 GM067945) to S.P.G. T.A.R. is a Howard Hughes Medical Institute Investigator. This article is subject to HHMI's Open Access to Publications policy. HHMI laboratory heads have previously granted a non-exclusive CC BY 4.0 license to the public and a sublicensable license to HHMI in their research articles. Pursuant to those licenses, the author-accepted manuscript of this article can be made freely available under a CC BY 4.0 license immediately upon publication.

## Author contributions

N.W. initiated the work on REEPs in autophagy. Y.S. identified Rop1 in fission yeast and performed initial experiments with *S. pombe*. N.W. performed all subsequent experiments with yeast Rop1 and the biochemical work. J.A.P. and S.P.G. performed quantitative mass spectrometry. T.A.R. supervised the project, and N.W., Y.S., and T.A.R. wrote the manuscript.

## Competing interests

The authors declare no competing interests.
