## [Peer review file · Nature Communications]

Reviewers' Comments:

Reviewer #1:

Remarks to the Author:

In this study, Wang et al. identified the REEP1 family protein Rop1 as a novel factor involved in macroautophagy in *Schizosaccharomyces pombe*. Autophagosome formation is impaired in cells lacking Rop1, and therefore these cells are defective in all of the macroautophagy-related pathways, including non-selective autophagy, ERphagy, mitophagy, and pexophagy. Rop1 has an amphipathic helix in the cytoplasmic C-terminal region, and this helix is required for membrane curvature generation *in vitro* and for autophagosome formation *in vivo*. In fluorescence microscopy, Rop1 is observed as puncta and upon autophagy induction it colocalizes with autophagy-related proteins including the phagophore marker Atg8. Moreover, based on the results of further fluorescent microscopy and electron microscopy with ascorbate peroxidase staining, the authors conclude that Rop1 localizes to the rim of the expanding phagophore. In addition, the authors show that mammalian REEP1 also localizes phagophores. Collectively, the authors propose the model that REEP1 proteins localize to and stabilize the rim of the phagophore to prevent phagophore swelling during autophagosome formation. I find this study provides an important insight into the molecular mechanism of membrane shaping during autophagosome formation, but the authors need to address the following issues to further support their conclusions and improve the manuscript.

1. Extended Data Fig. 1h: Although the authors stated that "deletion of Yop1 (*yop1Δ*) had no effect on ERphagy", the data show that Rtn1-GFP degradation is defective in *yop1Δ* cells.
2. Fig. 2: The authors should also quantify the colocalization of Atg8 with Atg1 and Atg9, especially for Atg1, because the Atg1 complex serves as a scaffold for autophagosome formation.
3. Sfb3 and Sec24 localize to not only the cytoplasm but also the ER and therefore are not adequate as cytosolic markers.
4. Although the authors attributed small vacuoles in *rop1Δ* cells to defective autophagy, this is unlikely because vacuolar size is normal in *atg8Δ* cells.
5. Fig. 2i: The authors should confirm that abnormal autophagic structures in *fsc1Δ rop1Δ* cells disappear by deletion of ATG8 to show that these are indeed autophagosome-related structures.
6. Fig. 3a-d: (i) The authors should state why they used recombinant sjRop1, instead of spRop1, for *in vitro* experiments. (ii) The authors should clarify the relationship between the mutants examined in Fig. 3c and d.
7. Fig. 3e and f: (i) The authors should show the localization of Rop1 mutants. (ii) The authors should also perform a cleavage assay of Hsc1-GFP to quantitatively estimate bulk autophagy activity in cells expressing Rop1 mutants.
8. Fig. 4d-i: The authors claimed that Rop1 localizes to the rim of expanding phagophores based on the results of fluorescence and electron microscopy, but the images are not clear enough. In fluorescent microscopy, Rop1 indeed localized to part of enlarged Atg8 signals, but it is unclear whether this represents Rop1 localization to the phagophore rim. Simultaneously observing the localization of Rop1, Atg8 (phagophore), and Atg2 (phagophore rim) in *ctl1Δ* cells in 3 colors would address this issue. The electron microscopy images of cells expressing Rop1-APEX2 are also unclear. A different method, such as immunoelectron microscopy, would be required.
9. Fig 4j-n: The authors showed that artificial Yop1 tethering to Atg2 rescued the growth defect on DTT-containing medium and the defect of autophagic activity of *rop1Δ* cells. (i) In this assay, the authors expected that Atg2-GBP and Yop1-GFP reside in the same phagophore membrane after artificial tethering. However, then, Yop1, which is an integral membrane protein, needs to be transferred from the ER to the phagophore, not an easily explainable process. It is also equally possible that Yop1-GFP retains in the ER membrane and functions to tether the ER to the phagophore via binding to Atg2-GBP in the phagophore rim. Even more critically, this analysis raised the possibility that Rop1 also functions on the ER side in ER-phagophore contact sites rather than in the phagophore

rim. The authors should more carefully consider this possibility and perform additional experiments to address this important issue related to the main conclusion of this work. (ii) The authors should also examine autophagic activity of yop1-GFP atg2-GBP cells by Hsc1-GFP degradation assay.

10. In analysis in mammalian cells, the authors should clarify whether REEP1 localizes to the rim of the phagophore by observing the localization of Reep1, LC3, and Atg2 in 3 colors. Alternatively, in some mammalian cell lines, phagophores can be observed in a clear "C" shape. These cell lines will be helpful for the authors to observe the phagophore rim localization of REEP1.

11. The authors should examine whether REEP1 is required for autophagosome formation in mammalian cells.

12. Page 8, line 241: "(Fig.4a, quantification in Fig.4b)" should be "(Fig.4a, quantification in Fig.4c)".

Reviewer #2:

Remarks to the Author:

This is an important paper showing that the yeast homolog of REEP1 (Rop1) is involved in autophagy generally (cytosolic, mitochondrial, peroxisomal), and in particular that Rop1 localizes to the growing phagophore rim. They demonstrate that REEP1, like the well-characterised Yop1, is able to curve membranes, and indeed that Yop1, if localized correctly to the phagophore, can rescue phagosome formation when Rop1 is missing.

They provide a model for Rop1's role in phagophore expansion and formation that is well supported by the data presented. Finally, they extend the results to human REEP1 and show evidence of the possibility that this mechanism might explain the association of REEP1 mutations with diseases such as hereditary spastic paraplegia, which is a compelling future direction.

The experimental designs and data quality are very high and there are multiple confirming pieces of evidence to support the authors' conclusions. The microscopy data is mostly quantified, which boosts confidence in the results and the interpretations. The paper is well written and clear.

The manuscript presents a very significant breakthrough in fundamental cell biology and is remarkable for the clarity and simplicity of the data.

Typo: Line 241, it should read 'quantification in Fig. 4c' (not 4b).

Reviewer #3:

Remarks to the Author:

In this interesting manuscript, Wang et al. propose that the membrane shaping protein Rop1 (the yeast orthologue of mammalian REEP1) is required for the formation of the phagophore rim. This conclusion is mainly based on the observation that the knock-down of Rop1 impairs ER-phagy, mitophagy, pexophagy as well as bulk autophagy, while this is not the case for the knockdown of Yop1 (the yeast orthologue of mammalian REEP5). The latter has been reported previously as an ER resident shaping protein. If correct, this study could be a real advance in our understanding of autophagy.

Although compelling I still have several concerns that this conclusion might be premature on the data presented in this study as outlined in the following.

REEP1 has been reported as an ER resident shaping protein, which interacts with many other ER resident proteins (e.g. Park et al, JCI 2010; Renvoise et al., HMG 2016). In support of this assumption, its knockout in mice caused alterations of the ER structure of cortical motoneurons (Beetz et al., JCI 2013) as well as in axons of ATL1/REEP1 double mutant mice (Zhu et al, HMG 2022). Actually, the current manuscript also shows that the ROP1-mNG construct labels the ER in yeast (extended data figure 6). Therefore, I wonder whether an altered structure of the ER may also result in a defect of the formation of the phagophore. Of course, a role as an ER shaping protein does not exclude another role for the formation of the phagophore rim. To my mind, however, the current

manuscript does not address properly this possible interconnection. Although the authors address the consequences of ROP1 deletion in yeast (extended data figure 3), I miss, how the ER might be affected upon disruption of YOP1/REEP5, which according to their concept is the prototype of an ER shaping protein within the protein family. Of course, a meaningful analysis also requires some kind of quantification. To my mind, similar analyses are also required for mammalian cells.

Another limitation of the study is that the authors do not provide any data on the localization of endogenous Rop1/REEP1, but instead rely on tagged overexpression, which may potentially cause artifacts. This is particularly relevant because some of the tags employed are even larger as the ROP1 protein as such.

More technically, I often miss sound quantifications and a statement on how often experiments have been repeated.

In the discussion, I miss how the current findings relate to previous reports on the function of REEP1 as e.g. for ER shaping and lipid droplet biogenesis. I lack an appropriate discussion taking into account that Rop1/REEP1 may potentially serve different functions.

In the following, I provide my concerns and suggestions for each figure point by point.

Figure 1: I cannot identify much of a difference between WT and the mutants upon treatment with DTT. In addition, a sound quantification would be required to evaluate the consequences of Rop1 deletion for mitophagy and bulk autophagy.

The MS data could be elaborated. Which proteins accumulate? Why is the effect much less severe for the Rop1 mutant compared to the Atg8 mutant, if all forms of autophagy are strongly impaired? Can the accumulation of selected proteins be confirmed by western blot? Can the authors perform an interactome of Rop1 because this may show some protein directly related to autophagy in support of their claim?

Figure 2: The studies related to the co-localization between Atg8 with either Atg1 or Atg9 also need to be quantified. For the quantification of AtG8 with Atg5 and Atg2 colocalization I miss statistics. In g, I would like to see the WT control for comparison. Again, I lack a quantification and statistics. In h and i I was not able to distinguish single and double membranes.

Figure 3: A control image showing pure liposomes would be helpful for comparison. Can the authors show that the small lipoparticles actually contain Rop1/REEP1 by immunogold labeling? I also wonder why the bulk of liposomes incubated with REEP1 were considerably larger in previous studies. Is this a difference between Rop1 and REEP1 or due the lipid composition? The authors furthermore claim that the shaping properties of human REEP1 are largely impaired for the L107P mutation and the deletion of aa 102-139 which correspond with the amphipathic helix. Because of the huge variability also shown in the exemplary images, a quantification is required.

Figure 4: The authors claim to show that Rop1 localizes to the phagophore rim. This conclusion is based on overexpression of a tagged constructs for Atg8 and Rop1. Actually, the overexpression of Rop1-mNG results in many labeled intracellular structures, but only a minor fraction colocalizes with Atg8 or Atg2 as judged from the images displayed. However, the quantification claims that 80% of Rop1 signals co-localize with Atg8 or Atg2. What is the reason for this discrepancy? The authors further report that Rop1 localizes to subregions of Atg8-positive structures. Here, the study needs to be strengthened by immunogold labeling and quantification. To my mind, the resolution of light microscopy does not allow any clear conclusion about the possible localization of Rop1 to subregions of the phagophore. The Rop1-APEX2 staining is a first step towards this direction but insufficient in the current state without quantification. I also wonder how the authors can distinguish expanding and closing phagophores in 2D TEM. The rescue experiment with Yop1 is elegant but raises the question whether it will also work with Rop1-GFP.

Figure 5: How does this punctate distribution of the overexpressed tagged REEP1 construct fit with a localization to the ER reported previously? The quantification of HA-signals of overexpressed REEP1-HA in extended data figure 7 is not clear to me. No signals co-localizing with ER? I am convinced that there is an overlap with the ER marker Sec61b. Do they get a clear overlap between REEP1-HA and the REEP1-Em to exclude artifacts by different tags? Actually the co-labeling shown in the inserts does not convince me. I cannot clearly distinguish the signal in phagophores from background noise. Did the authors check co-localization with ATG2? I would also like to see the ratio of REEP1-signals that do not colocalize with markers of autophagy. To state that the REEP1 signal in d is asymmetrical compared to LC3 signals is at best bold. Related to figure 5 and extended data figure 7, to which extent the functions of REEP1, REEP2, REEP3, and REEP4 differ from each other? Are they redundant? Do they show different expression patterns at the tissue level? Which paralogues are expressed in U2OS cells?

Reviewer #4:

Remarks to the Author:

Summary:

The authors in Wang et al. studied the initial formation of the early autophagophore closure, an event that is from biophysicochemical perspective, unfavorable and requires supportive proteins. Which those proteins are and how the high curvature of the phagophore rim is stabilized remains unknown. The authors utilize a plethora of methods and models to tackle this problem and to support their claims. While the title and abstract suggest a broader study the authors primarily work with *S. Pombe*. They prove that *rop1*, an ortholog of human REEP1, is involved and critical for proper autophagosome formation. In the following they verify the involvement of *rop1* in autophagy, its localizes to the forming autophagophore which seems to localize to the terminals of the closure. Accordingly, *rop1* deficient *S.pombe* is having problems forming mature autophagosomes and seemingly, problems forming defined early closures as demonstrated via EM. In the following, they perform a limited number of experiments also in human cells focusing on the human ortholog REEP1.

Major points:

1. Confirmation of knockouts: Why is the absence of the proteins of interest not demonstrated? I am not from the *S. Pombe* field, but this might be a critical missing control? I suggest doing this via PRM, in case suitable antibodies are missing.
2. I find it tricky to generalize because the experiments were primarily conducted in *S. Pombe*.
3. Regarding the function of mammalian REEP1. It might be that *rop1*'s functions and localizations are conserved for REEP1. However, for generalization, as the title suggests, knock out or knock down experiments in human cells confirming the findings in *S.Pombe* are required. This might be RNAi or more elegantly via endogenous e.g. FLAG-FKBPf36v tagging, allowing inducible depletion or pull-down of REEP1. Via Flag-tag, you could further perform AP-MS. Co-enrichment with autophagosomal proteins is elegant to confirm co-localization and maybe you learn more about the interactome are the membrane terminals. At a minimum, visualization via EM, confirming the disruption of autophagosome formation and quantification thereof, would be required control vs. depletion, ideally quantified and normalized to the total number of autophagosomes per cell. Currently, I would limit those findings to *S.Pombe*. If you want to generalize these findings, I want to see these or similar confirmative experiments.
4. Generally, the manuscript lacks quantifications. There are a lot of indications and correlations supporting certain claims, but without minimum required replicates done/mentioned (often missing), image analysis and statistics, I am having problems being convinced.
5. Figure 1h and proteomics experiments: The method described for those experiments needs improvement. Your peers will be unable to determine precisely what was done in the existing form. Generally, it is insufficient only to cite the previous descriptions of methods from another manuscript to satisfy proper R3 standards. You are welcome to reference those former manuscripts, but I request detailed descriptions here. Important missing minimum details are: 1. Which machine was used 2. Which mode was measured 3. Which software was used for the search and against which reference proteome (with date)? 4. Which exact statistics were used?
 - a. Please work on the display, highlight & name the top 5 up and down regulated proteins.
 - i. How do you explain the reduction in the levels of specific proteins?
 - b. Underlying Data Table S3: Either describe the processing in detail and provide the original data, or provide a table with all the analysis: Fold Change, p-Value and info if this was adjusted (corrected) and how.
 - c. Further, while understanding your normalization, I cannot track the samples/experiment in this data with the current labeling. Please have this tracible i.e. name the replicates intuitively.

Minor points:

- Ext Figure 8f: Misses numbering of close-ups

We thank the reviewers for their constructive criticism which we think has led to a greatly improved paper. We were happy to see that all reviewers found our paper interesting and potentially suitable for publication in Nature Communications. The revision has taken us some time, as we performed a large number of additional experiments requested by the reviewers.

The major changes in the manuscript are:

1. We agree with the reviewers that the imaging data do not have the sufficient resolution to claim that Rop1 localizes to the phagophore rim. However, we show that Rop1 colocalizes with Atg2, an autophagy component that has been reported to localize to the rim. Moreover, we show that this colocalization is required for Rop1's function in autophagy. We provide further data to support both points. Nevertheless, we agree with reviewer #1 that it is possible that Rop1 localizes instead to a specialized ER region, as EM tomography data show that high curvature ER domains are in close proximity to phagophore rims (Bieber et al. 2022; Li et al., 2023). We therefore rewrote the paper to acknowledge both possibilities.
2. We deleted the data on mammalian cells (previous Fig. 5a-d). These data were criticized as rather preliminary by three of the four reviewers, with which we agree, and we followed reviewer's #4 suggestion to focus the paper exclusively on *S. pombe*. As mentioned in our previous letter, we cannot easily test the function of REEP1 proteins in mammalian cells, as there are four REEP1-like proteins that are highly similar to each other, and at least three are expressed in any given cell line. The omission of the mammalian data does not affect any conclusion of the paper, and we believe that concentrating solely on the yeast experiments has led to more streamlined and stronger manuscript.

Another reason to focus our paper on *S. pombe* is that a competing paper has just been submitted as a preprint to Biorxiv. This paper also reports on the function of Rop1 in *S. pombe* (called Yep1 by the authors) and cites our paper (which we submitted to Biorxiv more than half a year ago). The two studies agree on most points, but there are some discrepancies, which we mention in the revised manuscript (p13). Given the competition, we would appreciate a rapid review of our manuscript.

Almost all figures of the revised manuscript contain new data. We now provide quantification for most experiments, added new mass spec results that include deletion of additional autophagy components, and provide more controls for the EM experiments with *fsc1Δ rop1Δ* cells. We now confirm that ER shaping proteins do not have a role in autophagy and use a biochemical assay to show that colocalization of Yop1 with Atg2 rescues the autophagy defect of the *rop1Δ* mutant. We also performed pull-down/mass spec experiments to look for interaction partners of Rop1. These results show that, despite its colocalization with Atg2, Rop1 does not physically interact with Atg2 (or other autophagy components). We had previously demonstrated that Rop1 plays a role in ERphagy, mitophagy, pexophagy, and bulk autophagy, and have now added a figure showing that it is also involved in autophagy of the Golgi (Fig. 1h). These data strengthen our claim that Rop1 is a general autophagy component. To further prove that Rop1 generates high membrane curvature, we extended the data in new Fig. 4 by showing

flotation experiments. These data confirm that these proteins form high-curvature lipoprotein particles. Because of the additional data, we split previous Fig. 1 into new Figs. 1 and 2, and previous Fig. 3 into new Figs. 4 and 5.

Below is a point-by-point response to the points raised by the reviewers (our response is in blue):

Reviewer #1 (Remarks to the Author):

In this study, Wang et al. identified the REEP1 family protein Rop1 as a novel factor involved in macroautophagy in *Schizosaccharomyces pombe*. Autophagosome formation is impaired in cells lacking Rop1, and therefore these cells are defective in all of the macroautophagy-related pathways, including non-selective autophagy, Erphagy, mitophagy, and pexophagy. Rop1 has an amphipathic helix in the cytoplasmic C-terminal region, and this helix is required for membrane curvature generation *in vitro* and for autophagosome formation *in vivo*. In fluorescence microscopy, Rop1 is observed as puncta and upon autophagy induction it colocalizes with autophagy-related proteins including the phagophore marker Atg8. Moreover, based on the results of further fluorescent microscopy and electron microscopy with ascorbate peroxidase staining, the authors conclude that Rop1 localizes to the rime of the expanding phagophore. In addition, the authors show that mammalian REEP1 also localizes phagophores. Collectively, the authors propose the model that REEP1 proteins localize to and stabilize the rim of the phagophore to prevent phagophore swelling during autophagosome formation. I find this study provides an important insight into the molecular mechanism of membrane shaping during autophagosome formation, but the authors need to address the following issues to further support their conclusions and improve the manuscript.

1. Extended Data Fig. 1h: Although the authors stated that “deletion of Yop1 (*yop1Δ*) had no effect on Erphagy”, the data show that Rtn1-GFP degradation is defective in *yop1Δ* cells.

We now use a general ER protein (Sec63-GFP) as a marker when testing the role of ER shaping proteins in autophagy (new Fig. 1e). Our results show that loss of Rop1 and Atg8, but not the ER shaping proteins Yop1, Rtn1, or Sey1, affect ERphagy. We also confirm that *rop1* deletion has no effect on ER morphology (Supplementary Fig. 2d). Together, these data show that Rop1 differs fundamentally from the ER shaping proteins.

2. Fig. 2: The authors should also quantify the colocalization of Atg8 with Atg1 and Atg9, especially for Atg1, because the Atg1 complex serves as a scaffold for autophagosome formation.

We now provide quantification for the colocalization of the various Atg proteins with Atg8 in both wild-type and *rop1* deletion strains (Fig. 3c-f).

3. Sfb3 and Sec24 localize to not only the cytoplasm but also the ER and therefore are not

adequate as cytosolic markers.

To follow bulk autophagy, we now use five different cytosolic markers: overexpressed YFP (Fig. 2a), Tdh1-mCh (Fig. 2b), Hsc1-GFP (Fig. 2c), Pyk1-mCh, and Pgc1-mCh (Supplementary Fig. 3a). As the reviewer pointed out and is now mentioned in the text, Sfb3 and Sec24 can be considered as markers for ERphagy and bulk autophagy (Supplementary Fig. 3b;c). We have also extended the mass spec data (Figs. 2d-g), which also largely report on bulk autophagy.

4. Although the authors attributed small vacuoles in *rop1Δ* cells to defective autophagy, this is unlikely because vacuolar size is normal in *atg8Δ* cells.

We agree with the reviewer that the small vacuole phenotype is probably not caused directly by an autophagy defect. We therefore have changed the text.

An interesting possibility is suggested by our pull-down experiments, where we found that a vacuolar ABC transporter is specifically associated with Rop1 under starvation conditions (see Supplementary Fig. 8c). Perhaps a loss of this interaction causes the small vacuole phenotype; however, we have refrained from discussing this possibility in the paper, as it is rather speculative.

5. Fig. 2i: The authors should confirm that abnormal autophagic structures in *fsc1Δ rop1Δ* cells disappear by deletion of *ATG8* to show that these are indeed autophagosome-related structures.

We now provide these data in Supplementary Fig. 5e;f and show the quantification of the experiment in Fig. 3i.

6. Fig. 3a-d: (i) The authors should state why they used recombinant sjRop1, instead of spRop1, for in vitro experiments. (ii) The authors should clarify the relationship between the mutants examined in Fig. 3c and d.

We used sjRop1 because spRop1 could not be purified in sufficient amounts (now mentioned in the text). However, the two proteins are very similar in sequence and our in vivo data with *S. japonicus* show that sjRop1 plays an analogous role as spRop1 (Supplementary Fig. 2e). We agree with the reviewer that it was confusing that we used different deletion mutants for Rop1 from the two different species. We therefore made two new mutants in sjRop1 ($\Delta 103-118$ and $\Delta 120-134$ in Fig. 4d) that are equivalents to the ones used for spRop1 (Fig. 5).

7. Fi. 3e and f: (i) The authors should show the localization of Rop1 mutants. (ii) The authors should also perform a cleavage assay of Hsc1-GFP to quantitatively estimate bulk autophagy activity in cells expressing Rop1 mutants.

We did test the localization of the Rop1 mutants but did not see much difference to the wild-type, except for one mutant. However, the wild-type shows a heterogeneous localization, as

the protein is found in punctae and possibly the ER and other locations (Fig. 6). The heterogeneous localization may mask changes in the mutants, and we therefore feel that these data are difficult to interpret and should not be included in the paper.

As requested by the reviewer, we have now performed the proteolytic cleavage assay with Hsc1-GFP and have quantified the effect of *rop1* deletion on Hsc1-GFP cleavage (Fig. 2c). We also used the assay to test various Rop1 mutations for their effect on autophagy (Fig. 5e).

8. Fig. 4d-i: The authors claimed that Rop1 localizes to the rim of expanding phagophores based on the results of fluorescence and electron microscopy, but the images are not clear enough. In fluorescent microscopy, Rop1 indeed localized to part of enlarged Atg8 signals, but it is unclear whether this represents Rop1 localization to the phagophore rim. Simultaneously observing the localization of Rop1, Atg8 (phagophore), and Atg2 (phagophore rim) in *ctl1Δ* cells in 3 colors would address this issue. The electron microscopy images of cells expressing Rop1-APEX2 are also unclear. A different method, such as immunoelectron microscopy, would be required.

Imaging Rop1, Atg8, and Atg2 simultaneously is challenging, particularly because all three proteins are expressed at low levels and a required far-red marker would be too dim. Instead, we now provide co-localization data for all possible pairwise combinations. These results show that Rop1 colocalizes with Atg2, and Atg2 localizes to a subdomain of Atg8-labeled punctae, as reported previously for starved *ctl1Δ* cells (Sun et al., 2013). Assuming that Atg2 localizes to the rim of phagophores as in budding yeast and other organisms, these data support our hypothesis that Rop1 localizes to the rim or to neighboring tips of ER tubules.

We now provide quantification for the APEX experiments (Fig. 6l). Although based on a small number of observed phagophores, there is a clear difference between the localization of Rop1 and Atg2 on the one hand, and Atg5 on the other, consistent with the assumption that Atg2 localizes to the rim and Atg5 to the entire phagophore.

We did try immunoelectron microscopy, but could not obtain convincing results, likely because the antibodies did not efficiently penetrate the sample after fixation. We believe that there is currently no way to further improve the localization data. We admit that we cannot make a strong point about Rop1's localization to the phagophore rim and mention this in the Discussion (p12). We now acknowledge that it is possible that Rop1 instead localizes to highly curved ER regions that are in close proximity to phagophore rims.

9. Fig 4j-n: The authors showed that artificial Yop1 tethering to Atg2 rescued the growth defect on DTT-containing medium and the defect of autophagic activity of *rop1Δ* cells. (i) In this assay, the authors expected that Atg2-GBP and Yop1-GFP reside in the same phagophore membrane after artificial tethering. However, then, Yop1, which is an integral membrane protein, needs to be transferred from the ER to the phagophore, not an easily explainable process. It is also equally possible that Yop1-GFP retains in the ER membrane and functions to tether the ER to the phagophore via binding to Atg2-GBP in the phagophore rim. Even more critically, this analysis raised the possibility that Rop1 also functions on the ER side in ER-phagophore contact

sites rather than in the phagophore rim. The authors should more carefully consider this possibility and perform additional experiments to address this important issue related to the main conclusion of this work. (ii) The authors should also examine autophagic activity of yop1-GFP atg2-GBP cells by Hsc1-GFP degradation assay.

We now use a cleavage assay with Tdh1-mCh to demonstrate that the expression of Yop1-GFP and Atg2-GBP rescues autophagy in a *rop1* deletion strain (new Fig. 7f). We also provide quantification for this experiment (Fig. 7g).

We agree with the reviewer that it is conceivable that Yop1-GFP remains in the ER and is dragged by its interaction with Atg2-GBP to contact sites between the ER and phagophores. We now acknowledge this possibility in the Discussion and the scheme of Fig. 7h. However, it should be noted that combining the bulk ER protein Sec63-GFP with Atg2-GBP does not rescue the autophagy defect of the *rop1* deletion (Fig. 7c;f). Thus, if Rop1 functions on the ER, it must localize to a specialized ER region. Based on EM tomography data, it seems possible that Rop1 sits at curved ER regions that are in immediate proximity of phagophore rims (Bieber et al. 2022; Li et al., 2023). One might therefore speculate that it could facilitate Atg2-mediated lipid transfer from the ER into phagophores. We now mention this possibility in the Discussion.

10. In analysis in mammalian cells, the authors should clarify whether REEP1 localizes to the rim of the phagophore by observing the localization of Reep1, LC3, and Atg2 in 3 colors. Alternatively, in some mammalian cell lines, phagophores can be observed in a clear "C" shape. These cell lines will be helpful for the authors to observe the phagophore rim localization of REEP1.

As mentioned above, we have decided to omit the experiments with mammalian cells. Although we attempted to find cell lines or genetic methods that lead to accumulation of unclosed phagophores for APEX-based EM studies, the results were not convincing.

11. The authors should examine whether REEP1 is required for autophagosome formation in mammalian cells.

Again, as explained above, we no longer present data with mammalian cells. We agree that these experiments are important and are constructing such lines; however, because of the presence of at least three REEP1 family proteins in every mammalian cell line, the experiments are not trivial.

12. Page 8, line 241: "(Fig.4a, quantification in Fig.4b)" should be "(Fig.4a, quantification in Fig.4c)".

The mistake was corrected.

Reviewer #2 (Remarks to the Author):

This is an important paper showing that the yeast homolog of REEP1 (Rop1) is involved in autophagy generally (cytosolic, mitochondrial, peroxisomal), and in particular that Rop1 localizes to the growing phagophore rim. They demonstrate that REEP1, like the well-characterised Yop1, is able to curve membranes, and indeed that Yop1, if localized correctly to the phagophore, can rescue phagosome formation when Rop1 is missing.

They provide a model for Rop1's role in phagophore expansion and formation that is well supported by the data presented. Finally, they extend the results to human REEP1 and show evidence of the possibility that this mechanism might explain the association of REEP1 mutations with diseases such as hereditary spastic paraplegia, which is a compelling future direction.

The experimental designs and data quality are very high and there are multiple confirming pieces of evidence to support the authors' conclusions. The microscopy data is mostly quantified, which boosts confidence in the results and the interpretations. The paper is well written and clear.

The manuscript presents a very significant breakthrough in fundamental cell biology and is remarkable for the clarity and simplicity of the data.

Typo: Line 241, it should read 'quantification in Fig. 4c' (not 4b).

The mistake was corrected.

Reviewer #3 (Remarks to the Author):

In this interesting manuscript, Wang et al. propose that the membrane shaping protein Rop1 (the yeast orthologue of mammalian REEP1) is required for the formation of the phagophore rim. This conclusion is mainly based on the observation that the knock-down of Rop1 impairs ER-phagy, mitophagy, pexophagy as well as bulk autophagy, while this is not the case for the knockdown of Yop1 (the yeast orthologue of mammalian REEP5). The latter has been reported previously as an ER resident shaping protein. If correct, this study could be a real advance in our understanding of autophagy.

Although compelling I still have several concerns that this conclusion might be premature on the data presented in this study as outlined in the following.

REEP1 has been reported as an ER resident shaping protein, which interacts with many other ER resident proteins (e.g. Park et al, JCI 2010; Renvoise et al., HMG 2016). In support of this assumption, its knockout in mice caused alterations of the ER structure of cortical motoneurons (Beetz et al., JCI 2013) as well as in axons of ATL1/REEP1 double mutant mice (Zhu et al, HMG 2022). Actually, the current manuscript also shows that the ROP1-mNG construct labels the ER

in yeast (extended data figure 6). Therefore, I wonder whether an altered structure of the ER may also result in a defect of the formation of the phagophore. Of course, a role as an ER shaping protein does not exclude another role for the formation of the phagophore rim. To my mind, however, the current manuscript does not address properly this possible interconnection. Although the authors address the consequences of ROP1 deletion in yeast (extended data figure 3), I miss, how the ER might be affected upon disruption of YOP1/REEP5, which according to their concept is the prototype of an ER shaping protein within the protein family. Of course, a meaningful analysis also requires some kind of quantification. To my mind, similar analyses are also required for mammalian cells.

We now show that only Rop1, and not the ER shaping proteins Yop1, Rtn1, or Sey1, affect ERphagy (Fig. 1e). We also now present Sec63 localization data for all relevant proteins (Supplementary Fig. 2d), which show that ER morphology is not greatly affected by single deletions (a slight defect is seen in *rtn1Δ* cells).

Endogenous Rop1 shows a punctate pattern, rather than the typical ER localization of Yop1 (Fig. 6a;b;e-g; Supplementary Fig. 7b). However, when Rop1 is overexpressed, it does label the bulk ER and colocalizes with Yop1 (this is the experiment the reviewer refers to, i.e. Supplementary Fig. 7c), suggesting that the punctate population originates from the ER. Taken together, our data show that there is a clear functional difference between Rop1 and Yop1.

We no longer present data on mammalian cells. We agree with the reviewer that more experiments are necessary to clarify why we observed vesicular localization vs ER localization of REEP1, as reported in the literature. These experiments also detracted from the major focus of the paper.

Another limitation of the study is that the authors do not provide any data on the localization of endogenous Rop1/REEP1, but instead rely on tagged overexpression, which may potentially cause artifacts. This is particularly relevant because some of the tags employed are even larger as the ROP1 protein as such.

This is a misunderstanding. We do present localization experiments with endogenous Rop1 expressed from the chromosomal locus under its own promoter (Fig. 6 and Supplementary Fig. 7).

More technically, I often miss sound quantifications and a statement on how often experiments have been repeated.

We now provide quantification of most experiments and give the number of cells analyzed.

In the discussion, I miss how the current findings relate to previous reports on the function of REEP1 as e.g. for ER shaping and lipid droplet biogenesis. I lack an appropriate discussion taking into account that Rop1/REEP1 may potentially serve different functions.

We now mention the previous results on the role of REEP1 proteins in microtubule interaction and lipid droplet formation (p13).

In the following, I provide my concerns and suggestions for each figure point by point.

Figure 1: I cannot identify much of a difference between WT and the mutants upon treatment with DTT. In addition, a sound quantification would be required to evaluate the consequences of Rop1 deletion for mitophagy and bulk autophagy.

DTT induces less ERphagy than starvation, leading to a weaker fluorescence signal in vacuoles (red arrowhead in Fig. 1b). DTT also causes aggregation of ER membrane proteins, which is visible as bright Yop1-tdT punctae (Fig. 1b). We now added asterisks to help the reader distinguish them from the dim vacuoles. Despite this complication, the difference between the vacuoles in wild-type and mutant cells should be obvious.

We now provide quantification for most experiments.

The MS data could be elaborated. Which proteins accumulate? Why is the effect much less severe for the Rop1 mutant compared to the Atg8 mutant, if all forms of autophagy are strongly impaired? Can the accumulation of selected proteins be confirmed by western blot?

We repeated the mass spec experiments and added *atg2Δ* cells as a further control (Fig. 2d-g). The results show that many proteins accumulating in *rop1Δ*, *atg2Δ*, or *atg8Δ* cells are identical. *rop1Δ* and *atg2Δ* cells have a less severe autophagy defect than *atg8Δ* cells.

We added additional panels in the figure reporting the mass spec data (Fig. 2d-g) and provide a detailed description of the methods employed.

Can the authors perform an interactome of Rop1 because this may show some protein directly related to autophagy in support of their claim?

We have now performed pull-down/mass spec experiments with Rop1 (Supplementary Fig. 8) using both starved cells and cells grown on rich medium. We used Yop1 as control. These data do not support a physical interaction between Rop1 and Atg2 (or other autophagy proteins). We therefore speculate that the colocalization of Rop1 and Atg2 may be caused indirectly, perhaps by Atg2 favoring high curvature membrane regions generated by Rop1 (p13).

Figure 2: The studies related to the co-localization between Atg8 with either Atg1 or Atg9 also need to be quantified. For the quantification of AtG8 with Atg5 and Atg2 colocalization I miss statistics.

We now provide quantification for all colocalizations (Fig. 3c-f). The statistical significance of differences is also given.

In g, I would like to see the WT control for comparison. Again, I lack a quantification and statistics. In h and i I was not able to distinguish single and double membranes.

A comparison of *fsc1Δ* cells with WT cells has been published (Sun et al., 2013) and we present the localization of Atg8 in WT cells in Fig. 3a. Quantification in panel g is unnecessary because all bright punctae disappear in the double mutant *fsc1Δ rop1Δ*. In addition, we now provide quantification for the EM images of the same cells in Fig. 3i.

As requested by reviewer #1, we added a control to the experiments in Fig. 3h, in which we deleted *atg8* in the *fsc1Δ* or *fsc1Δ rop1Δ* cells.

In most cases, the two membranes in autophagosomes are very close to one another. We now show an example where the two membranes can be seen (Fig. 3j). We also show a double membrane structure for cells treated with DTT (Supplementary Fig. 5h). It has been a general observation in the field that the two membranes of autophagosomes are often too close to be distinguished by ordinary EM (Nguyen et al., 2017).

Figure 3: A control image showing pure liposomes would be helpful for comparison. Can the authors show that the small lipoparticles actually contain Rop1/REEP1 by immunogold labeling? I also wonder why the bulk of liposomes incubated with REEP1 were considerably larger in previous studies. Is this a difference between Rop1 and REEP1 or due the lipid composition?

There are no previous studies that showed curvature-generation by REEP1. All previous work was done with the REEP5 subfamily.

Mutant REEP proteins, i.e. sjYop1(Δ142-165) and sjRop1(Δ107-120), provide better controls than empty liposomes, as the purified proteins contain detergent that can affect the reconstitutions. We have now quantified the diameters of the reconstituted liposomes observed with the different protein compositions (Fig. 4e).

Although Rop1 particles generated after expression in *E. coli* have a larger diameter than REEP5 particles (new quantification in Fig. 4c), they both are lipoprotein particles. This is now demonstrated by flotation experiments: whereas WT Rop1 stays near the bottom of the gradient, mutant Rop1 floats, as it generates less curvature and therefore localizes to large vesicles (new Fig. 4f).

The authors furthermore claim that the shaping properties of human REEP1 are largely impaired for the L107P mutation and the deletion of aa 102-139 which correspond with the amphipathic helix. Because of the huge variability also shown in the exemplary images, a quantification is required.

We now provide quantification (Fig. 4e).

Figure 4: The authors claim to show that Rop1 localizes to the phagophore rim. This conclusion is based on overexpression of a tagged constructs for Atg8 and Rop1.

Again, this is a misunderstanding, as we present localization data for endogenous Rop1, Atg8, and Atg2, all expressed under their own promoters from their native genomic loci (Fig. 6). Only the experiment in Fig. 6a uses overexpressed Atg8.

Actually, the overexpression of Rop1-mNG results in many labeled intracellular structures, but only a minor fraction colocalizes with Atg8 or Atg2 as judged from the images displayed. However, the quantification claims that 80% of Rop1 signals co-localize with Atg8 or Atg2. What is the reason for this discrepancy?

The experiment shown in Fig. 6 was performed with endogenous levels of Rop1-mNG. The text may have been misleading. We meant to say that ~80% of all Atg2 or Atg8 punctae contain Rop1. We have therefore changed the labels in Fig. 6c and phrased the text unambiguously. As pointed out by the reviewer, only a small fraction of Rop1 co-localizes with Atg proteins upon starvation.

The authors further report that Rop1 localizes to subregions of Atg8-positive structures. Here, the study needs to be strengthened by immunogold labeling and quantification. To my mind, the resolution of light microscopy does not allow any clear conclusion about the possible localization of Rop1 to subregions of the phagophore.

We provide a magnified view of the co-localization of Atg2 and Rop1 (Fig. 6g). Previous results showed that Atg2 localizes to a subdomain of Atg8-labeled punctae in starved *ct/1Δ* cells (Sun et al., 2013). Assuming that Atg2 localizes to the rim of phagophores as in budding yeast and other organisms, these data support our hypothesis that Rop1 is close to the rim. We agree with the reviewer that immuno EM would be desirable, but we have been unable to obtain convincing data. We now concede that the rim localization of both Atg2 and Rop1 requires further experimentation. In the Discussion, we mention two models, one in which Rop1 stabilizes the high curvature of the phagophore rim, and another in which Rop1 stabilizes the curvature of specialized ER regions and facilitates Atg2-mediated lipid flow from the ER into phagophores.

The Rop1-APEX2 staining is a first step towards this direction but insufficient in the current state without quantification. I also wonder how the authors can distinguish expanding and closing phagophores in 2D TEM.

We now provide quantification for the APEX experiments (Fig. 6l). Although based on a small number of observed phagophores, there is a clear difference between the localization of Rop1 and Atg2 on the one hand, and Atg5 on the other.

We now call phagophores "open" if they have a C-shape and "closing" when they are circular, but still labeled with Rop1 or Atg2.

The rescue experiment with Yop1 is elegant but raises the question whether it will also work with Rop1-GFP.

We made a GFP fusion of Rop1, but it turned out to be not fully functional, unlike the other fusions of Rop1 used in the paper. The combination of Rop1-GFP and Atg2-GBP is therefore not a good control. We attach the data for the information of the reviewer.

Legend to the figure: The indicated fusions of Rop1 or Atg2 were expressed from the endogenous genomic locus in wild-type (wt) cells. The cells were grown for 3 days in DTT and plated after serial dilution. Note that Rop1-GFP and Rop1-tdTomato were not fully functional and were therefore not used in our study.

Figure 5: How does this punctate distribution of the overexpressed tagged REEP1 construct fit with a localization to the ER reported previously? The quantification of HA-signals of overexpressed REEP1-HA in extended data figure 7 is not clear to me. No signals co-localizing with ER? I am convinced that there is an overlap with the ER marker Sec61b. Do they get a clear overlap between REEP1-HA and the REEP1-Em to exclude artifacts by different tags?

As mentioned above, we have decided to omit the experiments with mammalian cells. We are performing experiments to clarify the discrepancy between the vesicular localization of REEP1 proteins observed in our experiments and the ER localization reported in the literature.

Actually the co-labeling shown in the inserts does not convince me. I cannot clearly distinguish the signal in phagophores from background noise. Did the authors check co-localization with ATG2? I would also like to see the ratio of REEP1-signals that do not colocalize with markers of autophagy. To state that the REEP1 signal in d is asymmetrical compared to LC3 signals is at best bold.

Related to figure 5 and extended data figure 7, to which extent the functions of REEP1, REEP2, REEP3, and REEP4 differ from each other? Are they redundant? Do they show different expression patterns at the tissue level? Which paralogues are expressed in U2OS cells?

Again, we have deleted all experiments with mammalian cells.

Reviewer #4 (Remarks to the Author):

Summary:

The authors in Wang et al. studied the initial formation of the early autophagophore closure, an event that is from biophysicochemical perspective, unfavorable and requires supportive proteins. Which those proteins are and how the high curvature of the phagophore rim is stabilized remains unknown. The authors utilize a plethora of methods and models to tackle this problem and to support their claims. While the title and abstract suggest a broader study the authors primarily work with *S. Pombe*. They prove that *rop1*, an ortholog of human REEP1, is involved and critical for proper autophagosome formation. In the following they verify the involvement of *rop1* in autophagy, its localizes to the forming autophagophore which seems to localize to the terminals of the closure. Accordingly, *rop1* deficient *S.pombe* is having problems forming mature autophagosomes and seemingly, problems forming defined early closures as demonstrated via EM. In the following, they perform a limited number of experiments also in human cells focusing on the human ortholog REEP1.

Major points:

1. Confirmation of knockouts: Why is the absence of the proteins of interest not demonstrated? I am not from the *S. Pombe* field, but this might be a critical missing control? I suggest doing this via PRM, in case suitable antibodies are missing.

All knock-outs were confirmed by PCR-based genotyping, as is standard in the field. The PCR products confirmed that all genes' coding sequences were replaced by inserted antibiotic expression cassettes. This is now mentioned in the Methods.

2. I find it tricky to generalize because the experiments were primarily conducted in *S. Pombe*.

We agree with the reviewer that all mechanistic insight is based on experiments with *S. pombe*. As explained above, we have deleted all experiments with mammalian cells. Accordingly, the text and title were changed.

3. Regarding the function of mammalian REEP1. It might be that *rop1*'s functions and localizations are conserved for REEP1. However, for generalization, as the title suggests, knock out or knock down experiments in human cells confirming the findings in *S.Pombe* are required. This might be RNAi or more elegantly via endogenous e.g. FLAG-FKBPf36v tagging, allowing inducible depletion or pull-down of REEP1. Via Flag-tag, you could further perform AP-MS. Co-enrichment with autophagosomal proteins is elegant to confirm co-localization and maybe you

learn more about the interactome are the membrane terminals. At a minimum, visualization via EM, confirming the disruption of autophagosome formation and quantification thereof, would be required control vs. depletion, ideally quantified and normalized to the total number of autophagosomes per cell. Currently, I would limit those findings to *S. Pombe*. If you want to generalize these findings, I want to see these or similar confirmative experiments.

As suggested by the reviewer, we now limit our findings to *S. pombe*.

4. Generally, the manuscript lacks quantifications. There are a lot of indications and correlations supporting certain claims, but without minimum required replicates done/mentioned (often missing), image analysis and statistics, I am having problems being convinced.

We now provide quantification for essentially all experiments.

5. Figure 1h and proteomics experiments: The method described for those experiments needs improvement. Your peers will be unable to determine precisely what was done in the existing form. Generally, it is insufficient only to cite the previous descriptions of methods from another manuscript to satisfy proper R3 standards. You are welcome to reference those former manuscripts, but I request detailed descriptions here. Important missing minimum details are: 1. Which machine was used 2. Which mode was measured 3. Which software was used for the search and against which reference proteome (with date)? 4. Which exact statistics were used?

We have now provided the requested information. The Methods section has been greatly expanded. The mass spec experiments were repeated with an additional control (*atg2Δ* cells; Fig. 2d-g). The results show that many proteins that are up- or down-regulated in *rop1Δ*, *atg2Δ*, or *atg8Δ* cells are identical (Fig. 2e and 2f). *rop1Δ* and *atg2Δ* cells have a less severe autophagy defect than *atg8Δ* cells.

- a. Please work on the display, highlight & name the top 5 up and down regulated proteins.
 - i. How do you explain the reduction in the levels of specific proteins?
- b. Underlying Data Table S3: Either describe the processing in detail and provide the original data, or provide a table with all the analysis: Fold Change, p-Value and info if this was adjusted (corrected) and how.
- c. Further, while understanding your normalization, I cannot track the samples/experiment in this data with the current labeling. Please have this traceable i.e. name the replicates intuitively.

We now provide a list of the proteins that are up- or down-regulated in the absence of Rop1, Atg8, or Atg2 (Fig. 2e-g). We also provide the requested information in Supplementary Table 3.

We are not sure why the levels of some proteins are reduced in the absence of autophagy components (Fig. 2f), but we now mention these results in the text. There may be a compensatory response to the increase of other proteins that are normally degraded by autophagy. Interestingly, the same proteins are downregulated in all three deletion mutants.

Minor points:

- Ext Figure 8f: Misses numbering of close-ups

The data have been deleted.

REVIEWERS' COMMENTS

Reviewer #1 (Remarks to the Author):

The authors have addressed most of the issues I raised for the original manuscript, but some concerns remain as follows.

1. Regarding their response to my comment #7, the localization of Rop1 mutants are important information for readers to consider the effects of the mutations and therefore should be included in the manuscript.
2. The authors added the results for degradation of the cis-Golgi protein Anp1 in the revised manuscript, but I understand that no literature reported autophagic degradation of the Golgi (this protein) in yeasts. The authors should check if Anp1 degradation depends on autophagy (Atg proteins) or remove the results as they are not essential for this work.
3. Fig. 6l: The label for the y axis should be "C-shaped phagophore (%)".

Reviewer #3 (Remarks to the Author):

The authors have addressed most of my concerns appropriately and the manuscript has improved considerably.

From the experiments presented, they provide multiple evidence using different approaches that ROP1 is involved in autophagy in yeast. Their idea that it helps to shape the phagophore rim is fascinating. I am still not absolutely convinced of the data that claim that ROP1 localizes to the rim of the autophagophore, which overlaps with ATG2, but I acknowledge the efforts of the authors to address my concerns and the way they discuss this limitation. The rescue experiments with the targeting to ATG2 is very elegant. As outlined by the authors it will need further studies to assess, whether REEP1 serves the same function in vertebrates including humans.

I just want to point to some minor issues:

In general, the arrows, arrowheads as well as stars are too small.

Fig. 1b,c: The quantification suggests that no vacuolar signal is present in delta rop1 or delta atg8.

The exemplary images, however, seem to have some residual vacuolar signals?

Fig. 3k: statistics?

Fig. 6e and g: For me this example is not very convincing, while I can follow the argumentation of the authors for d and f.

Fig. S2: I guess the labeling of S2 e and g were mixed up.

Fig 5c: I recommend to label the blot: IB Atg8.

Reviewer #4 (Remarks to the Author):

Dear authors,

The comments of the reviewers, including mine, were in a very similar direction and you did a good job in implementing these comments, as it appears to me. I appreciate the implementation of relevant statistical quantifications and the adjustment of appropriate claims based on these findings. Utmost appreciated is your greatly improved method part and the representation of the data in an understandable/interpretable way. Regarding this, please make sure the underlying MS runs are available via ProteomeXchange. Further, I am still missing info about the S. Pombe proteome version your runs were mapped to, this needs to be indicated/made available. You can also provide that file in the supplement, also fine for me. Overall, the manuscript was greatly improved.

Minor comments:

- Figure 1b) Label "DTT 24h" seems to be written in different font size within the label
- Line 891: "analyzed" to imaged
- Figure 1c) Suggest removing the minor ticks on y-axis and have 25, 50, 75% indicated. Suggest removing bar borders and change error-bars to black color.
- Figure 2b) Suggest removing the minor ticks on y-axis.
- Figure 4a) "Strepaavidin" to Streptavidin
- Figure 4c) "Purificated" to purified. Further, error bars are very hard to see, I suggest adjusting this for better visibility.
- Figure 5c) Some text overlaps with figure
- Figure S6d) "Strepaavidin" to Streptavidin
- Figure S6e) "Strepaavidin" to Streptavidin

I suggest changing the line thickness of all displays to the same. Also, I am not entirely sure if you used the same font across all figures. This should be double-checked.

I suggest revising the figure legends once more with a focus on shortening them while keeping all necessary information. Further, in case across the figure the same statistical confidence intervals and indications were used, it is sufficient to indicate that at the end of the legend once, applicable for all analyses.

Point-by-point response to the reviewers' comments:

REVIEWERS' COMMENTS

Reviewer #1 (Remarks to the Author):

The authors have addressed most of the issues I raised for the original manuscript, but some concerns remain as follows.

1. Regarding their response to my comment #7, the localization of Rop1 mutants are important information for readers to consider the effects of the mutations and therefore should be included in the manuscript.

We have now included the localization of the Rop1 mutants in Fig. S7d. The localization of the mutants is now mentioned on p. 9.

2. The authors added the results for degradation of the cis-Golgi protein Anp1 in the revised manuscript, but I understand that no literature reported autophagic degradation of the Golgi (this protein) in yeasts. The authors should check if Anp1 degradation depends on autophagy (Atg proteins) or remove the results as they are not essential for this work.

As suggested by the reviewer, we have removed the data on Anp1.

3. Fig. 6I: The label for the y axis should be "C-shaped phagophore (%)".

The figure was apparently misleading. We have now split Fig. 6I into two parts to make clear that we first determined the percentage of cells with C-shaped phagophores (left part) and then the percentage of these cells that contain the indicated proteins at the rim of C-shaped phagophores (right part).

Reviewer #3 (Remarks to the Author):

The authors have addressed most of my concerns appropriately and the manuscript has improved considerably.

From the experiments presented, they provide multiples evidence using different approaches that ROP1 is involved in autophagy in yeast. They idea that it helps to shpae the phagophore rim is fascinating. I am still not abosultely convinced of the data that claim that ROP1 localizes to the rim of the autophagophore, which overlaps with ATG2, but I acknowledge the efforts of the authors to address my concerns and the way they discuss this limitation. The rescue experiments with the targeting to ATG2 is very elegant. As outlined by the authors it will need further studies to assess, whether REEP1 serves the same function in vertebrates including humans.

I just want to point to some minor issues:

In general, the arrows, arrowheads as well as stars are too small.

We have increased the size of the symbols.

Fig. 1b,c: The quantification suggests that no vacuolar signal is present in delta rop1 or delta atg8. The exemplary images, however, seem to have some residual vacuolar signals?

The images are consistent with the quantification. We have added a sentence to the text (p. 3) to explain that the bright punctae in Fig. 1b are caused by DTT-induced membrane protein aggregation.

Fig. 3k: statistics?

We have added the statistics in the figure and the Source Data file.

Fig. 6e and g: For me this example is not very convincing, while I can follow the argumentation of the authors for d and f.

Rop1 localizes not only to a sub-domain of Atg8 that contains Atg2, but also to other punctae. So, not all Rop1 punctae co-localize with Atg2.

Fig. S2: I guess the labeling of S2 e and g were mixed up.

This was corrected.

Fig 5c: I recommend to label the blot: IB Atg8.

The reviewer is probably talking about Fig. S5c. However, this is a Coomassie-stained gel, not a blot. We have added an arrow and label on the side to indicate the position of Atg8.

Reviewer #4 (Remarks to the Author):

Dear authors,

The comments of the reviewers, including mine, were in a very similar direction and you did a good job in implementing these comments, as it appears to me. I appreciate the implementation of relevant statistical quantifications and the adjustment of appropriate claims based on these findings. Utmost appreciated is your greatly improved method part and the representation of the data in an understandable/interpretable way. Regarding this, please make sure the underlying MS runs are available via ProteomeXchange.

The data have been deposited to the ProteomeXchange Consortium via the PRIDE partner repository with the dataset identifier PXD043260. This is now mentioned under "Data availability".

Further, I am still missing info about the S.Pombe proteome version your runs were mapped to, this needs to be indicated/made available. You can also provide that file in the supplement, also fine for me.

The mass spec data were mapped to the following S. pombe database: <https://www.uniprot.org/proteomes/UP000002485>. This is now mentioned in the Methods.

Overall, the manuscript was greatly improved.

Minor comments:

- Figure 1b) Label "DTT 24h" seems to be written in different font size within the label

Changed, as requested.

- Line 891: "analyzed" to imaged

Changed, as requested.

- Figure 1c) Suggest removing the minor ticks on y-axis and have 25, 50, 75% indicated. Suggest removing bar borders and change error-bars to black color.

Changed, as suggested.

- Figure 2b) Suggest removing the minor ticks on y-axis.

Changed, as suggested.

- Figure 4a) "Strepaavidin" to Streptavidin

Changed, as requested.

- Figure 4c) "Purificated" to purified. Further, error bars are very hard to see, I suggest adjusting this for better visibility.

Changed, as requested.

- Figure 5c) Some text overlaps with figure

The reviewer is probably referring to Fig. 5d. We have placed "stationary cells" above the figure.

- Figure S6d) "Streptavidin" to Streptavidin
- Figure S6e) "Streptavidin" to Streptavidin

This was corrected.

I suggest changing the line thickness of all displays to the same. Also, I am not entirely sure if you used the same font across all figures. This should be double-checked.

We adjusted the line thickness and fonts in all displays.

I suggest revising the figure legends once more with a focus on shortening them while keeping all necessary information. Further, in case across the figure the same statistical confidence intervals and indications were used, it is sufficient to indicate that at the end of the legend once, applicable for all analyses.

We added the statistics information to the legends, as requested by the reviewer and the editors.